# Understanding Gradient Clipping in Private SGD: A Geometric Perspective

**Xiangyi Chen**
University of Minnesota
chen5719@umn.edu

**Zhiwei Steven Wu**
Carnegie Mellon University
zstevenwu@cmu.edu

**Mingyi Hong**
University of Minnesota
mhong@umn.edu

## Abstract

Deep learning models are increasingly popular in many machine learning applications where the training data may contain sensitive information. To provide formal and rigorous privacy guarantee, many learning systems now incorporate differential privacy by training their models with *(differentially) private SGD*. A key step in each private SGD update is *gradient clipping* that shrinks the gradient of an individual example whenever its $\ell_2$ norm exceeds some threshold. We first demonstrate how gradient clipping can prevent SGD from converging to a stationary point. We then provide a theoretical analysis that fully quantifies the clipping bias on convergence with a disparity measure between the gradient distribution and a geometrically symmetric distribution. Our empirical evaluation further suggests that the gradient distributions along the trajectory of private SGD indeed exhibit symmetric structure that favors convergence. Together, our results provide an explanation why private SGD with gradient clipping remains effective in practice despite its potential clipping bias. Finally, we develop a new perturbation-based technique that can provably correct the clipping bias even for instances with highly asymmetric gradient distributions.

## 1 Introduction

Many modern applications of machine learning rely on datasets that may contain sensitive personal information, including medical records, browsing history, and geographic locations. To protect the private information of individual citizens, many machine learning systems now train their models subject to the constraint of differential privacy [Dwork et al., 2006], which informally requires that no individual training example has a significant influence on the trained model. To achieve this formal privacy guarantee, one of the most popular training methods, especially for deep learning, is *differentially private stochastic gradient descent* (DP-SGD) [Bassily et al., 2014, Abadi et al., 2016b]. At a high level, DP-SGD is a simple modification of SGD that makes each step differentially private with the *Gaussian mechanism*: at each iteration $t$, it first computes a gradient estimate $g_t$ based on a random subsample, and then updates the model using a noisy gradient $\tilde{g}_t = g_t + \eta$, where $\eta$ is a noise vector drawn from a multivariate Gaussian distribution.

Despite the simple form of DP-SGD, there is a major disparity between its theoretical analysis and practical implementation. The formal privacy guarantee of Gaussian mechanism requires that the per-coordinate standard deviation of the noise vector $\eta$ scales linearly with the $\ell_2$ sensitivity of the gradient estimate $g_t$—that is, the maximal change on $g_t$ in $\ell_2$ distance if by changining a single example. To bound the $\ell_2$-sensitivity, existing theoretical analyses typically assume that the loss function is $L$-Lipschitz in the model parameters, and the constant $L$ is known to the algorithm designer for setting the noise rate [Bassily et al., 2014, Wang and Xu, 2019]. Since this assumption implies that the gradient of each example has $\ell_2$ norm bounded by $L$, any gradient estimate from averaging over the gradients of $m$ examples has $\ell_2$-sensitivity bounded by $L/m$. However, in many

practical settings, especially those with deep learning models, such Lipschitz constant or gradient bounds are not a-priori known or even computable (since it involves taking the worst case over both examples and pairs of parameters). In practice, the bounded $\ell_2$-sensitivity is ensured by *gradient clipping* [Abadi et al., 2016b] that shrinks an individual gradient whenever its $\ell_2$ norm exceeds certain threshold $c$. More formally, given any gradient $g$ on a simple example and a clipping threshold $c$, the gradient clipping does the following

$$\text{clip}(g, c) = g \cdot \min\left(1, \frac{c}{\|g\|}\right). \tag{1}$$

However, the clipping operation can create a substantial bias in the update direction. To illustrate this clipping bias, consider the following two optimization problems even without the privacy constraint.

**Example 1.** Consider optimizing $f(x) = \frac{1}{3}\sum_{i=1}^{3} \frac{1}{2}(x - a_i)^2$ over $x \in \mathbb{R}$, where $a_1 = a_2 = -3$ and $a_3 = 9$. Since the gradient $\nabla f(x) = x - 1$, the optimum is $x^* = 1$. Now suppose we run SGD with gradient clipping with a threshold of $c = 1$. At the optimum, the gradients for all three examples are clipped and the expected clipped gradient is $1/3$, which leads the parameter to move away from $x^*$.

**Example 2.** Let $f(x) = \frac{1}{2}\sum_{i=1}^{2} \frac{1}{2}(x - a_i)^2$, where $a_1 = -3$ and $a_2 = 3$. The minimum of $f$ is achieved at $x^* = 0$, where the expected clipped gradient is also 0. However, SGD with clipped gradients and $c = 1$ may never converge to $x^*$ since the expected clipped gradients are all 0 for any $x \in [-2, 2]$, which means all these points are "stationary" for the algorithm.

Both examples above show that clipping bias can prevent convergence in the worst case. Existing analyses on gradient clipping quantify this clipping bias either with 1) the difference between clipped and unclipped gradients [Pichapati et al., 2019], or 2) the fraction of examples with gradient norms exceeding the clip threshold $c$ [Zhang et al., 2019]. These approaches suggest that a small clip threshold will lead to large clipping bias and worsen the training performance of DP-SGD. However, in practice, DP-SGD often remains effective even with a small clip threshold, which indicates a gap in the current theoretical understanding of gradient clipping.

## 1.1 Our results

We study the effects of gradient clipping on SGD and DP-SGD and provide:

**Symmetricity-based analysis.** We characterize the clipping bias on the convergence to stationary points through the geometric structure of the gradient distribution. To isolate the clipping effects, we first analyze the non-private SGD with gradient clipping (but without Gaussian perturbation), with the following key analysis steps. We first show that the inner product $\mathbb{E}[\langle \nabla f(x_t), g_t \rangle]$ goes to zero in SGD, where $\nabla f(x)$ denotes the true gradient and $g_t$ denotes a clipped stochastic gradient. Secondly, we show that when the gradient distribution is symmetric, the inner product $\mathbb{E}[\langle \nabla f(x_t), g_t \rangle]$ upper bounds a constant re-scaling of $\|\nabla f(x_t)\|$, and so SGD minimizes the gradient norm. We then quantify the clipping bias via a coupling between the gradient distribution and a nearby symmetric distribution and express it as a disparity measure (that resembles the Wasserstein distance) between the two distributions. As a result, when the gradient distributions are near-symmetric or when the clipping bias favors convergence, the clipped gradient remains aligned with the true gradient, even if clipping aggressively shrinks almost all the sample gradients.

**Theoretical and empirical evaluation of DP-SGD.** Building on the SGD analysis, we obtain a similar convergence guarantee on DP-SGD with gradient clipping. Importantly, we are able to prove such convergence guarantee even *without* Lipschitzness of the loss function, which is often required for DP-SGD analyses. We also provide extensive empirical studies to investigate the gradient distributions of DP-SGD across different epochs on two real datasets. To visualize the symmetricity of the gradient distributions, we perform multiple random projections on the gradients and examine the two-dimensional projected distributions. Our results suggest that the gradient distributions in DP-SGD quickly exhibit symmetricity, despite the asymmetricity at initialization.

**Gradient correction mechanism.** Finally, we provide a simple modification to DP-SGD that can mitigate the clipping bias. We show that perturbing the gradients *before* clipping can provably reduce the clipping bias for any gradient distribution. The pre-clipping perturbation does not by itself provide privacy guarantees, but can trade-off the clipping bias with higher variance.

## 1.2 Related work

The divergence caused by the clipping bias was also studied by prior work. In Pichapati et al. [2019], an adaptive gradient clipping method is analyzed and the divergence is characterized by a bias depending on the difference between the clipped and unclipped gradients. However, they study a different variant of clipping that bounds the $\ell_\infty$ norm of the gradient instead of $\ell_2$ norm; the latter, which we study in this paper, is the more commonly used clipping operation [Abadi et al., 2016b,a]. In Zhang et al. [2019], the divergence is characterized by a bias depending on the clipping probability. These results suggest that, the clipping probability as well as the bias are inversely proportional to the size of the clipping threshold. For example, small clipping threshold results in large bias in the gradient estimation, which can potentially lead to worse training and generalization performance. Thakkar et al. [2019] provides another adaptive gradient clipping heuristic that sets the threshold based on a privately estimated quantile, which can be viewed as minimizing the clipping probability. In a very recent work, Song et al. [2020] shows that gradient clipping can lead to constant regret in worst case and it is equivalent to huberizing the loss function for generalized linear problems. Compared with the aforementioned works, our result shows that the bias caused by gradient clipping can be 0 when the gradient distribution is symmetric, revealing the effect of gradient distribution beyond clipping probabilities.

## 2  Convergence of SGD with clipped gradient

In this section, we analyze convergence of SGD with clipped gradient, but without the Gaussian perturbation. This simplification is useful for isolating the clipping bias. Consider the standard stochastic optimization formulation

$$\min_x f(x) \triangleq \mathbb{E}_{s \sim D}[f(x, s)] \tag{2}$$

where $D$ denotes the underlying distribution over the examples $s$. In the next section, we will instantiate $D$ as the empirical distribution over the private dataset. We assume that the algorithm is given access to a stochastic gradient oracle: given any iterate $x_t$ of SGD, the oracle returns $\nabla f(x_t) + \xi_t$, where $\xi_t$ is independent noise with zero mean. In addition, we assume $f(x)$ is G-smooth, i.e. $\|\nabla f(x) - \nabla f(y)\| \le G\|x - y\|, \forall x, y$. At each iteration $t$, SGD with gradient clipping performs the following update:

$$x_{t+1} = x_t - \alpha \text{clip}(\nabla f(x_t) + \xi_t, c) := x_t - \alpha g_t. \tag{3}$$

where $g_t = \text{clip}(\nabla f(x_t) + \xi_t, c)$ denotes the realized clipped gradient.

To carry out the analysis of iteration (3), we first note that the standard convergence analysis for SGD-type method consists of two main steps:

**S1)** Show that the following key term diminishes to zero: $\mathbb{E}[\langle \nabla f(x_t), g_t \rangle]$.

**S2)** Show that the aforementioned quantity is proportional to $\|\nabla f(x_t)\|^2$ or $c\|\nabla f(x_t)\|$, indicating that the size of gradient also decreases to zero.

In our analysis below, we will see that showing the first step is relatively easy, while the main challenge is to show that the second step holds true. Our first result is given below.

**Theorem 1.** *Let $G$ be the Lipschitz constant of $\nabla f$ such that $\|\nabla f(x) - \nabla f(y)\| \le G\|x - y\|, \forall x, y$. For SGD with gradient clipping of threshold $c$, if we set $\alpha = \frac{1}{\sqrt{T}}$, we have*

$$\frac{1}{T} \sum_{t=1}^{T} \mathbb{E}\left[\langle \nabla f(x_t), g_t \rangle\right] \le \frac{D_f}{\sqrt{T}} + \frac{G}{2\sqrt{T}} c^2 \tag{4}$$

*where $D_f = f(x_1) - \min_x f(x)$.*

Note that for SGD without clipping, we have $\mathbb{E}[\langle \nabla f(x_t), g_t \rangle] = \|\nabla f(x_t)\|^2$, so the convergence can be readily established. However, when clipping is applied, the expectation is different but if we have $\mathbb{E}[\langle \nabla f(x_t), g_t \rangle]$ being positive or scaling with $\|\nabla f(x_t)\|$, we can still establish a convergence guarantee. However, the divergence examples (Example 1 and 2) indicate proving this second step requires additional conditions. Now we study a geometric condition that is observed empirically.

## 2.1 Symmetricity-Based Analysis on Gradient Distribution

Let $p_t(\xi_t)$ be the probability density function of $\xi_t$ and $\tilde{p}_t(\xi_t)$ is an arbitrary distribution. To quantify the clipping bias, we start the analysis with the following decomposition:

$$\mathbb{E}_{\xi_t \sim p}[\langle \nabla f(x_t), g_t \rangle] = \mathbb{E}_{\xi_t \sim \tilde{p}}[\langle \nabla f(x_t), g_t \rangle] + \underbrace{\int \langle \nabla f(x_t), \text{clip}(\nabla f(x_t) + \xi_t, c) \rangle (p_t(\xi_t) - \tilde{p}_t(\xi_t)) d\xi_t}_{b_t} \quad (5)$$

Recall that $\xi_t \sim p$ in (5) is the gradient noise caused by data sampling, we can choose $\tilde{p}(\xi_t)$ to be some "nice" distribution that can effectively relate $\mathbb{E}_{\xi_t \sim \tilde{p}}[\langle \nabla f(x_t), g_t \rangle]$ to $\|\nabla f(x_t)\|^2$ and the remaining term will be treated as the bias. This way of splitting ensures that when the gradients follow a "nice" distribution, the bias will diminish with the distance between $p$ and $\tilde{p}_t$. More precisely, we want to find a distribution $\tilde{p}$ such that $\mathbb{E}_{\xi_t \sim \tilde{p}}[\langle \nabla f(x_t), g_t \rangle]$ is lower bounded by norm squared of the true gradient and thus convergence can be ensured.

A straightforward "nice" distribution will be one that can ensure $\langle \nabla f(x_t), g_t \rangle \geq \Omega(\|\nabla f(x_t)\|_2^2), \forall g_t$, i.e. all stochastic gradients are positively aligned with the true gradient. This may be satisfied when the gradient is large and the noise $\xi$ is bounded. However, when the gradient is small, it is hard to argue that this can still be true in general. Specifically, in the training of neural nets, the cosine similarities between many stochastic gradients and the true gradient (i.e. $\cos(\nabla f(x_t), \nabla f(x_t) + \xi_t)$) can be negative, which implies that this assumption does not hold (see Figure 3 in Section 4).

Although Figure 3 seems to exclude the *ideal* distribution, we observe that the distribution of cosine of the gradients appears to be *symmetric*. Will such a "symmetricity" property help define the "nice" distribution for gradient clipping? If so, how to characterizes the performance of gradient clipping in this situation? In the following result, we rigorously answer to these questions.

**Theorem 2.** *Assume $\tilde{p}(\xi_t) = \tilde{p}(-\xi_t)$, gradient clipping with threshold $c$ has the following properties.*

*1. If $\|\nabla f(x_t)\| \leq \frac{3}{4}c$, then* $\mathbb{E}_{\xi_t \sim \tilde{p}}[\langle \nabla f(x_t), g_t \rangle] \geq \|\nabla f(x_t)\|^2 \mathbb{P}_{\xi_t \sim \tilde{p}}\left(\|\xi_t\| < \frac{c}{4}\right)$

*2. If $\|\nabla f(x_t)\| > \frac{3}{4}c$, then* $\mathbb{E}_{\xi_t \sim \tilde{p}}[\langle \nabla f(x_t), g_t \rangle] \geq \frac{3}{4}c\|\nabla f(x_t)\| \mathbb{P}_{\xi_t \sim \tilde{p}}\left(\|\xi_t\| < \frac{c}{4}\right)$

Theorem 2 states that when the noise distribution is symmetric, gradient clipping will keep the expected clipped gradients positively aligned with the true gradient. This is the desired property that can guarantee convergence. The probability term characterizes the possible slow down caused by gradient clipping, we provide more discussions and experiments on this term in the Appendix.

Combining Theorem 2 with Theorem 1, we have Corollary 1 to fully characterize the convergence behavior of SGD with gradient clipping.

**Corollary 1.** *Consider the SGD algorithm with gradient clipping given in (3). Set $\alpha = \frac{1}{\sqrt{T}}$, and choose $\tilde{p}_t(\xi) = \tilde{p}_t(-\xi)$. Then the following holds:*

$$\frac{1}{T}\sum_{t=1}^{T} \mathbb{P}_{\xi_t \sim \tilde{p}_t}\left(\|\xi_t\| < \frac{c}{4}\right) \min\left\{\|\nabla f(x_t)\|, \frac{3}{4}c\right\}\|\nabla f(x_t)\| \leq \frac{D_f}{\sqrt{T}} + \frac{G}{2\sqrt{T}}c^2 - \frac{1}{T}\sum_{t=1}^{T} b_t, \quad (6)$$

*where we have defined $b_t := \int \langle \nabla f(x_t), \text{clip}(\nabla f(x_t) + \xi_t, c) \rangle (p_t(\xi_t) - \tilde{p}_t(\xi_t)) d\xi_t$.*

Therefore, as long as the probabilities $\mathbb{P}_{\xi_t \sim \tilde{p}_t}\left(\|\xi_t\| < \frac{c}{4}\right)$ are bounded away from 0 and the symmetric distributions $\tilde{p}_t$ are close approximations to $p_t$ (small bias $b_t$),[1] then gradient norm goes to 0. Moreover, when $\|\xi_t\|$ is drawn from a sub-gaussian distribution with constant variance, the probability does not diminish with the dimension. This is consistent with the observations in recent work of Li et al. [2020], Gur-Ari et al. [2018] on deep learning training, and we also provide our own empirical evaluation on the probability term in the Appendix.

Note if the bias is negative and very large, the bound on the rhs will not be meaningful. Therefore, it is useful to further study properties of such bias term. In the next section, we will discuss how large the bias term can be for a few choices of $p$ and $\tilde{p}$. It turns out that the accumulation of $b_t$ can be negative and can help in some cases.

## 2.2 Beyond symmetric distributions

Theorem 2 and Corollary 1 suggest that as long as the distribution $p$ is sufficiently close to a symmetric distribution $\tilde{p}$, the convergence bias expressed as $\sum_{t=1}^{T} b_t$ will be small. We now show that our bias decomposition result enables us to analyze the effect of the bias even for some highly asymmetric distributions. Note that when $b_t \geq 0$, the bias in fact helps convergence according Corollary 1. We now provide three examples where $b_t$ can be non-negative. Therefore, near-symmetricity is not a necessary condition for convergence, our symmetricity-based analysis remains an effective tool to establish convergence for a broad class of distributions.

**Positively skewed.** Suppose $p$ is positively skewed, that is, $p(\xi) \geq p(-\xi)$, for all $\xi$ with $\langle \xi, \nabla f(x) \rangle > 0$. With such distributions, the stochastic gradients tend to be positively aligned with the true gradient. If one chooses $\tilde{p}(\xi_t) = \frac{1}{2}(p(\xi_t) + p(-\xi_t))$, the bias $b_t$ can be written as

$$\int_{\xi_t \in \{\xi : \langle \xi, \nabla f(x_t) \rangle > 0\}} \langle \nabla f(x_t), \text{clip}(\nabla f(x_t) + \xi_t, c) - \text{clip}(\nabla f(x_t) - \xi_t, c) \rangle (\frac{1}{2}(p(\xi_t) - p(-\xi_t))) d\xi_t,$$

which is always positive since $\langle \nabla f(x_t), \text{clip}(\nabla f(x_t) + \xi_t, c) - \text{clip}(\nabla f(x_t) - \xi_t, c) \rangle \geq 0$. Substituting into (5), we have $\mathbb{E}_{\xi_t \sim p}[\langle \nabla f(x_t), g_t \rangle]$ strictly larger than $\mathbb{E}_{\xi_t \sim \tilde{p}}[\langle \nabla f(x_t), g_t \rangle]$, which means the positive skewness help convergence (we want $\mathbb{E}_{\xi_t \sim p}[\langle \nabla f(x_t), g_t \rangle]$ as large as possible).

**Mixture of symmetric.** The distribution of stochastic gradient $\nabla f(x_t) + \xi_t$ is a mixture of two symmetric distributions $p_0$ and $p_1$ with mean 0 and $v$ respectively. Such a distribution might be possible when most of samples are well classified. In this case, even though the distribution of $\xi_t$ is not symmetric, one can apply similar argument of Theorem 2 to the component with mean $v$, and the zero mean component yield a bias 0. In particular, let $w_0$ be the probability that $\nabla f(x_t) + \xi_t$ is drawn from $p_0$. One can choose $\tilde{p} = p - w_0 p_0$ which is the component symmetric over $v$. The bias become

$$\int \langle \nabla f(x_t), \text{clip}(\nabla f(x_t) + \xi_t, c) \rangle w_0 p_0(\xi_t) d\xi_t = 0 \tag{7}$$

since $p_0(\xi_t)$ corresponds to a zero mean symmetric distribution of $\nabla f(x_t) + \xi_t$. Note that despite $\tilde{p} = p - w_0 p_0$ is not a distribution since $\int \tilde{p}(\xi_t) = 1 - w_0$, Theorem 2 can still be applied with everything on RHS of inequalities multiplied by $1 - w_0$ because one can apply Theorem 2 to distribution $\tilde{p}(\xi_t)/(1 - w_0)$ and then scale everything down.

**Mixture of symmetric or positively skewed.** If $p$ is a mixture of multiple symmetric or positively skewed distributions, one can split the distributions into multiple ones and use their individual properties. E.g. one can easily establish convergence guarantee for $p$ being a mixture of $m$ spherical distributions with mean $u_1, ..., u_m$ and $\langle f(x_t), u_i \rangle \geq 0, \forall i \in [m]$ as in the following theorem.

**Theorem 3.** *Given $m$ distributions with the pdf of the $i$th distribution being $p_i(\xi) = \phi_i(\|\xi - u_i\|)$ for some function $\phi_i$. If $\nabla f(x_t) = \sum_{i=1}^{n} w_i u_i$ for some $w_i \geq 0, \sum_{i=1}^{m} w_i = 1$. Let $p'(\xi) = \sum_{i=1}^{m} w_i p_i(\xi - \nabla f(x_t))$, be a mixture of these distributions with zero mean. If $\langle u_i, \nabla f(x_t) \rangle \geq 0, \forall i \in [m]$, we have*

$$\mathbb{E}_{\xi_t \sim p'}[\langle \nabla f(x_t), g_t \rangle] \geq \|\nabla f(x_t)\| \sum_{i=1}^{m} w_i \min(\|u_i\|, \frac{3}{4}c) \cos(\nabla f(x_t), u_i) \mathbb{P}_{\xi_t \sim p_i}(\|\xi_t\| < \frac{c}{4}) \geq 0$$

Besides these examples of favorable biases above, there are also many cases where $b_t$ can be negative and lead to a convergence gap, such as negatively skewed distributions or multimodal distributions with highly imbalanced modes. We have illustrated possible distributions in our divergence examples (Examples 1 and 2). In such cases, one should expect that clipping has an adversarial impact on the convergence guarantee. However, as we also show in Section 4, the gradient distributions on real datasets tend to be symmetric, and so their clipping bias to be small.

## 3 DP-SGD with Gradient Clipping

We now extend the results above to analyze the overall convergence DP-SGD with gradient clipping. To match up with the setting in Section 2, we consider the distribution $D$ to be the empirical distribution over a private dataset $S$ of $n$ examples $\{s_1, \ldots, s_n\}$, and so $f(x) = \frac{1}{n} \sum_{i=1}^{n} f(x, s_i)$.

For any iterate $x_t \in \mathbb{R}^d$ and example $s_i$, let $\xi_{t,i} = \nabla f(x_t, s_i) - \nabla f(x_t)$ denote the gradient noise on the example, and $p_t$ denote the distribution over $\xi_{t,i}$. At each iteration $t$, DP-SGD performs:

$$x_{t+1} = x_t - \alpha \left( \left( \frac{1}{|S_t|} \sum_{i \in S_t} \text{clip}(\nabla f(x_t) + \xi_{t,i}, c) \right) + Z_t \right) \quad (8)$$

where $S_t$ is a random subsample of $S$ (with replacement)[2] and $Z_t \sim \mathcal{N}(0, \sigma^2 I)$ is the noise added for privacy. We first recall the privacy guarantee of the algorithm below:

**Theorem 4** (Privacy (Theorem 1 in Abadi et al. [2016b])). *There exist constants $u$ and $v$ so that given the number of iterations $T$, for any $\epsilon \leq uq^2 T$, where $q = \frac{|S_t|}{n}$, DP-SGD with gradient clipping of threshold $c$ is $(\epsilon, \delta)$-differentially private for any $\delta > 0$, if $\sigma^2 \geq v \frac{c^2 T \ln\left(\frac{1}{\delta}\right)}{n^2 \epsilon^2}$.*

By accounting for the sub-sampling noise and Gaussian perturbation in DP-SGD, we obtain the following convergence guarantee, where we further bound the clipping bias term $b_t$ with the Wasserstein distance between the gradient distribution and a coupling symmetric distribution.

**Theorem 5** (Convergence). *Let $d$ be the dimensionality of the parameters. For DP-SGD with gradient clipping, if we set $\alpha = \frac{\sqrt{D_f d \ln(\frac{1}{\delta})}}{n \epsilon c \sqrt{L}}$, $\tilde{p}_t(\xi_t) = \tilde{p}_t(-\xi_t)$, let $m = |S_t|$, there exist $u$ and $v$ such that for any $\epsilon \leq u \frac{m^2}{n^2} T$, $\sigma^2 = v \frac{c^2 T \ln\left(\frac{1}{\delta}\right)}{n^2 \epsilon^2}$, we have*

$$\frac{1}{T} \sum_{t=1}^{T} \mathbb{P}_{\xi_t \sim \tilde{p}}\left( \|\xi_t\| < \frac{c}{4} \right) h_c(\|\nabla f(x_t)\|) \leq \left( \frac{1}{2}v + \frac{3}{2} \right) c \frac{\sqrt{D_f G d \ln(\frac{1}{\delta})}}{n \epsilon} + \frac{1}{T} \sum_{t=1}^{T} W_{\nabla f(x_t), c}(\tilde{p}_t, p_t)$$

*where $h_c(y) = \min(y^2, \frac{3}{4}cy)$ and $W_{v,c}(p, p')$ is the Wasserstein distance between $p$ and $p'$ with metric function $d_{v,c}(a, b) = |\langle v, \text{clip}(v + a, c)\rangle - \langle v, \text{clip}(v + b, c)\rangle|$ and $D_f \geq f(x_1) - \min_x f(x)$.*

**Remark on convergence rate.** DP-SGD achieves convergence rate of $O(\sqrt{d}/(n\epsilon))$ in the existing literature. As shown in Theorem 5, with gradient clipping, the rate becomes $O(\sqrt{d}/(n\epsilon))$+clipping bias. When gradient distribution is symmetric, the convergence rate can recover to $O(\sqrt{d}/(n\epsilon))$. This result implies that when gradient distribution is symmetric, the clipping operation will only affect the convergence rate of DP-SGD by a constant factor. In addition, since the clipping bias is the Wasserstein distance between the empirical gradient distribution and an oracle symmetric distribution, it can be small when the gradient distribution is approximate symmetric.

**Remark on the Wasserstein distance.** In (6), it is clear that the convergence bias $b_t$ can be bounded by the total variation distance between $p_t$ and $\tilde{p}_t$. However, this bound becomes trivial when $p_t$ is the empirical distribution over a finite sample, because the total variation distance is always 2 when $\tilde{p}$ is continuous. In addition, the bias is hard to interpret when without further transformation. This is why we bound $b_t$ by the Wasserstein distance as follows:

$$-b_t = \int \langle \nabla f(x_t), \text{clip}(\nabla f(x_t) + \xi_t, c)\rangle (\tilde{p}(\xi_t) - p(\xi_t)) d\xi_t$$

$$= \int \langle \nabla f(x_t), \text{clip}(\nabla f(x_t) + \xi_t, c)\rangle \tilde{p}(\xi_t) d\xi_t - \int \langle \nabla f(x_t), \text{clip}(\nabla f(x_t) + \xi'_t, c)\rangle p(\xi'_t) d\xi'_t$$

$$= \int \int (\langle \nabla f(x_t), \text{clip}(\nabla f(x_t) + \xi_t, c)\rangle - \langle \nabla f(x_t), \text{clip}(\nabla f(x_t) + \xi'_t, c)\rangle) \gamma(\xi_t, \xi'_t) d\xi_t d\xi'_t$$

$$\leq \int \int |\langle \nabla f(x_t), \text{clip}(\nabla f(x_t) + \xi_t, c)\rangle - \langle \nabla f(x_t), \text{clip}(\nabla f(x_t) + \xi'_t, c)\rangle| \gamma(\xi_t, \xi'_t) d\xi_t d\xi'_t \quad (9)$$

where $\gamma$ is any joint distribution with marginal $\tilde{p}$ and $p$. Thus, we have

$$b_t \leq \inf_{\gamma \in \Gamma(\tilde{p}, p)} \int \int |\langle \nabla f(x_t), \text{clip}(\nabla f(x_t) + \xi_t, c)\rangle - \langle \nabla f(x_t), \text{clip}(\nabla f(x_t) + \xi'_t, c)\rangle| \gamma(\xi_t, \xi'_t) d\xi_t d\xi'_t$$

where $\Gamma(\tilde{p}, p)$ is the set of all couplings with marginals $\tilde{p}$ and $p$ on the two factors, respectively. If we define the distance function $d_{y,c}(a, b) = |\langle y, \text{clip}(y + a, c)\rangle - \langle y, \text{clip}(y + b, c)\rangle|$, we have

$$b_t \leq \inf_{\gamma \in \Gamma(\tilde{p}, p)} \int \int d_{\nabla f(x_t), c}(\xi_t, \xi'_t) \gamma(\xi_t, \xi'_t) d\xi_t d\xi'_t \quad (10)$$

which is Wasserstein distance defined on the distance function $d_{\nabla f(x_t),c}$ and it converges to the distance between the population distribution of gradient and $\tilde{p}$ with $n$ being large. Thus, if the population distribution of gradient is approximate symmetric, the bias term tends to be small. In addition, the distance function is uniformly bounded by $\|\nabla f(x)\|c$ which makes it is more favorable than $\ell_2$ distance. Compared with the expression of $b_t$ in Corollary 1, the Wasserstein distance is easier to interpret when $\tilde{p}$ is discrete.

## 4    Experiments

In this section, we investigate whether the gradient distributions of DP-SGD are approximate symmetric in practice. However, since the gradient distributions are high-dimensional, certifying symmetricity is in general intractable. We instead consider two simple proxy measures and visualizations.

**Setup.** We run DP-SGD implemented in Tensorflow [3] on two popular datasets MNIST [LeCun et al., 2010] and CIFAR-10 [Krizhevsky et al., 2009]. For MNIST, we train a CNN with two convolution layers with 16 4×4 kernels followed by a fully connected layer with 32 nodes. We use DP-SGD to train the model with $\alpha = 0.15$, and a batchsize of 128. For CIFAR-10, we train a CNN with two convolutional layers with 2×2 max pooling of stride 2 followed by a fully connected layer, all using ReLU activation, each layer uses a dropout rate of 0.5. The two convolution layer has 32 and 64 3×3 kernels, the fully connected layer has 1500 nodes. We use $\alpha = 0.001$ and decrease it by 10 times every 20 epochs. The clip norm of both experiments is set to be $c = 1$ and the noise multiplier is 1.1.

**Visualization with random projections.** We visualize the gradient distribution by projecting the gradient to a two-dimensional space using random Gaussian matrices. Note that given any symmetric distribution, its two-dimensional projection remains symmetric for any projection matrix. On the contrary, if for all projection matrix, the projected gradient distribution is symmetric, the original gradient distribution should also be symmetric. We repeat the projection using different randomly generated matrices and visualize the induced distributions.

From Figure 1, we can see that on both datasets, the gradient distribution is non-symmetric before training (Epoch 0), but over the epochs, the gradient distributions become increasingly symmetric. The distribution of gradients on MNIST at the end of epoch 9 projected to a random two-dimensional space using different random matrices is shown in Figure 2. It can be seen that the approximate symmetric property holds for all 8 realizations. We provide many more visualizations from different realized random projections across different epochs in the Appendix.

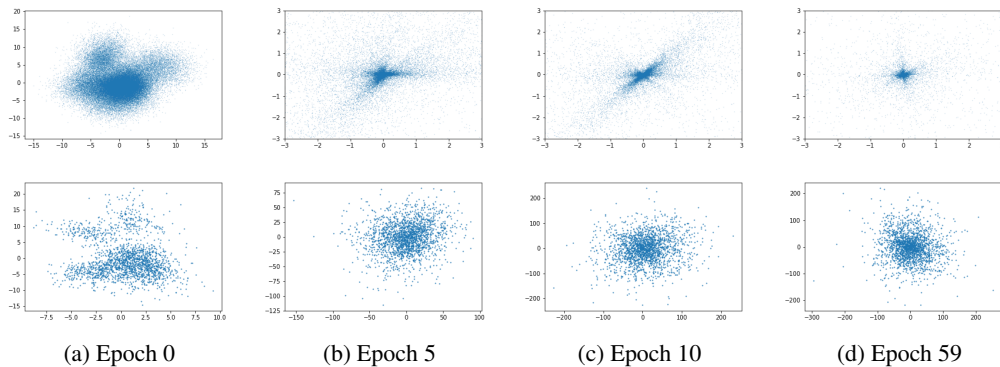

| (a) Epoch 0 | (b) Epoch 5 | (c) Epoch 10 | (d) Epoch 59 |

Figure 1: Gradient distributions on MNIST (top row) and CIFAR-10 (bottom row) at the end of different epochs (indexed by columns). The gradients for epoch 0 are computed at initialization (before training).

**Symmetricity of angles.** We also measure the cosine similarities between per-sample stochastic gradients and the true gradient. We observe that the cosine similarities between per-sample stochastic gradients and the true gradient (i.e. $\cos(\nabla f(x_t) + \xi_{t,i}, \nabla f(x_t))$) is approximate symmetric around 0 as shown in the histograms in Figure 3.

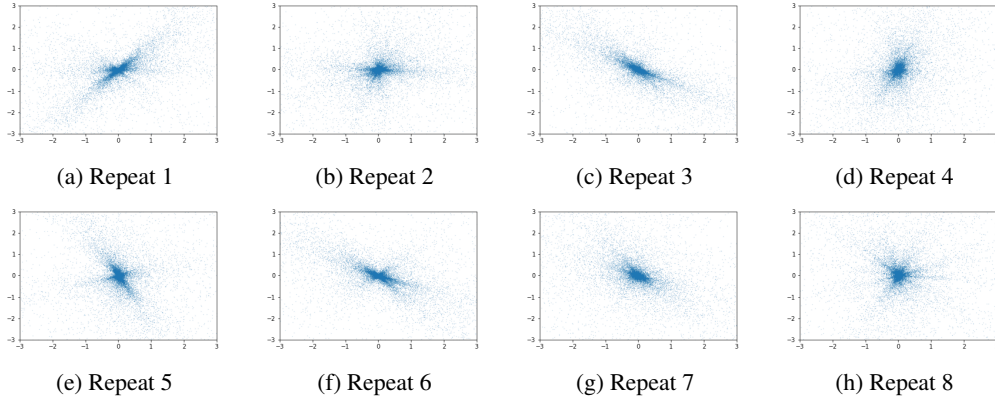

(a) Repeat 1        (b) Repeat 2        (c) Repeat 3        (d) Repeat 4

(e) Repeat 5        (f) Repeat 6        (g) Repeat 7        (h) Repeat 8

Figure 2: Gradient distributions on MNIST at the end of epoch 9 projected using different random matrices.

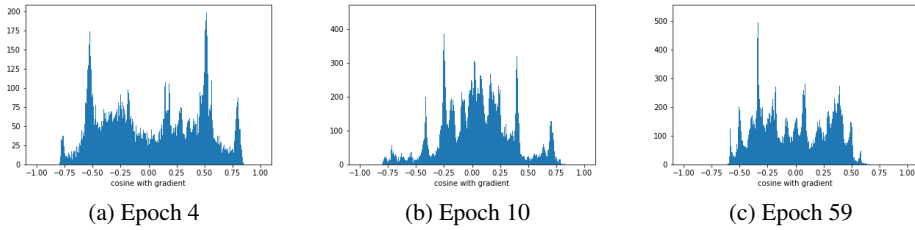

(a) Epoch 4                (b) Epoch 10                (c) Epoch 59

Figure 3: Histogram of cosine between stochastic gradients and the true gradient at the end of different epochs.

## 5 Mitigating Clipping Bias with Perturbation

From previous analyses, SGD with gradient clipping and DP-SGD have good convergence performance when the gradient noise distribution is approximately symmetric or when the gradient bias favors convergence (e.g., mixture of symmetric distributions with aligned mean). Although in practice, gradient distributions do exhibit (approximate) symmetry (see Sec. 4), it would be desirable to have tools to handle situations where the clipping bias does not favor convergence. Now we provide an approach to decrease the bias. If one adds some Gaussian noise before clipping, i.e.

$$g_t = \text{clip}(\nabla f(x_t) + \xi_t + k\zeta_t, c), \zeta_t \sim \mathcal{N}(0, I), \tag{11}$$

we can prove $|b_t| = O\left(\frac{\sigma_{\xi_t}^2}{k^2}\right)$ as in Theorem 6.

**Theorem 6.** *Let* $g_t = clip(\nabla f(x_t) + \xi_t + k\zeta_t, c)$ *and* $\zeta_t \sim \mathcal{N}(0, I)$. *Then gradient clipping algorithm has following properties:*

$$\mathbb{E}_{\xi_t \sim p, \zeta_t}[\langle \nabla f(x_t), g_t \rangle] \geq \|\nabla f(x_t)\| \min\left\{\|\nabla f(x_t)\|, \frac{3}{4}c\right\} \mathbb{P}(\|k\zeta_t\| < \frac{c}{4}) - O(\frac{\sigma_{\xi_t}^2}{k^2}) \tag{12}$$

*where* $\sigma_{\xi_t}^2$ *is the variance of the gradient noise* $\xi_t$.

More discussion can be found in the Appendix. Note that when the perturbation approach is applied to DP-SGD, the update rule (8) becomes

$$x_{t+1} = x_t - \alpha\left(\left(\frac{1}{|S_t|}\sum_{i \in S_t}\text{clip}(\nabla f(x_t) + \xi_{t,i} + k\zeta_{t,i}, c)\right) + Z_t\right),$$

where each per-sample stochastic gradient is be perturbed by the noise. By adding the noise, one trade-offs bias with variance. Larger noise make the algorithm converges possibly slower but better. This trick can be helpful when the gradient distribution is not favorable. To verify its effect in practice, we run DP-SGD with gradient clipping on a few unfavorable problems including examples in Section 1 and a new high dimensional example. We set $\sigma = 1$ on all the examples (i.e. $Z_t \sim \mathcal{N}(0, I)$). For the new example, we minimize the function $f(x) = \frac{1}{n}\sum_{i=1}^{n}\frac{1}{2}\|x - z_i\|^2$ with $n = 10000$. Each $z_i$

is drawn from a mixture of isotropic Gaussian with 3 components of dimension 10. The covariance matrix of all components is $I$ and the means of the 3 components are drawn from $\mathcal{N}(0, 36I)$, $\mathcal{N}(0, 4I)$, $\mathcal{N}(0, I)$, respectively. We set $\alpha = 0.015$ for the new examples and $\alpha = 0.001$ for the examples in Section 1. Figure 5 shows $\|x_t - \arg\min_x f(x)\|$ versus $t$. We can see DP-SGD with gradient clipping converges to non-optimal points as predicted by theory. In contrast, pre-clipping perturbation ensures convergence.

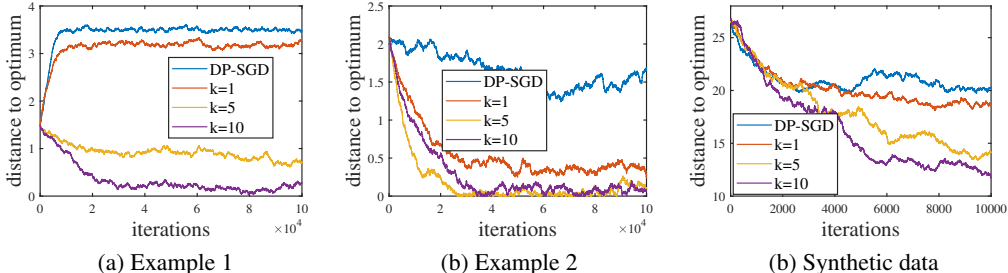

(a) Example 1  (b) Example 2  (b) Synthetic data

Figure 4: Effect of pre-clipping perturbation on the examples.

# 6   Conclusion and Future Work

In this paper, we provide a theoretical analysis on the effect of gradient clipping in SGD and private SGD. We provide a new way to quantify the clipping bias by coupling the gradient distribution with a geometrically symmetric distribution. Combined with our empirical evaluation showing that gradient distribution of private SGD follows some symmetric structure along the trajectory, these results provide an explanation why gradient clipping works in practice. We also provide a perturbation-based technique to reduce the clipping bias even for adversarial instances.

There are some interesting directions for future work. One of the main message of this paper is that when gradient distribution is symmetric, gradient clipping will not be detrimental to the performance of DP-SGD. Thus, looking for methods to symmetrify the gradient distribution could be an interesting topic. Another interesting direction is to study gradient distribution of different types of models empirically. We notice the gradient distribution of CNNs on MNIST and CIFAR-10 might be symmetric and a clipping threshold around 1 works well. However, McMahan et al. [2017] found a relatively large clipping threshold around 10 works best for LSTMs. This implies the gradient distribution on some models might be less symmetric and a concrete empirical analysis on it could motivate future research. Finally, it could be interesting to investigate properties of gradient clipping on a broader class of gradient distributions beyond symmetricity.

# 7   Broader Impacts

This paper aims to bridge the theory and practice of a commonly used privacy-preserving learning algorithm—differentially private SGD (DP-SGD). Our results provide theoretical understandings on the gradient clipping effects on the convergence behavior of the algorithm, which can further inform practical tuning of this important hyperparameter. As deep learning on sensitive data becomes increasingly common, our work provides theoretical insights for practitioners to perform reliable privacy-preserving machine learning.

## Acknowledgments and Disclosure of Funding

The research is supported in part by a NSF grant CMMI-1727757, a Google Faculty Research Award, a J.P. Morgan Faculty Award, and a Facebook Research Award.

## Footnotes

[1]Both Theorem 2 and Corollary 1 hold under a more relaxed condition of $\tilde{p}(\xi) = \tilde{p}(-\xi)$ for $\xi$ with $\ell_2$ norm exceeding $c/4$.

[2]Alternatively, subsampling with replacement [Wang et al., 2019] and Poisson subsampling [Zhu and Wang, 2019] have also been proposed.

[3]https://github.com/tensorflow/privacy/tree/master/tutorials

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
