[Supplementary Material]

# A  Proof of Theorem 1

By smoothness assumption, we have

$$f(x_{t+1}) \leq f(x_t) + \langle \nabla f(x_t), x_{t+1} - x_t \rangle + \frac{1}{2} G \|x_{t+1} - x_t\|^2. \tag{13}$$

Then, by update rule and the fact that $\|g_t\| \leq c$, we have

$$\begin{aligned} f(x_{t+1}) \leq &f(x_t) - \alpha \langle \nabla f(x_t), g_t \rangle + \frac{1}{2} G \alpha^2 \|g_t\|^2 \\ \leq &f(x_t) - \alpha \langle \nabla f(x_t), g_t \rangle + \frac{1}{2} G \alpha^2 c^2. \end{aligned} \tag{14}$$

Take expectation, sum over $t$ from 1 to $T$, divide both sides by $T\alpha$, rearranging and substituting into $\alpha = \frac{1}{\sqrt{T}}$, we get

$$\begin{aligned} \frac{1}{T} \sum_{t=1}^{T} \mathbb{E}[\langle \nabla f(x_t), g_t \rangle] \leq &\frac{1}{T\alpha}(f(x_1) - f(x_{T+1})) + \frac{1}{2} G\alpha c^2 \\ \leq &\frac{1}{\sqrt{T}} \mathbb{E}[f(x_1) - f(x_{T+1})] + \frac{1}{2\sqrt{T}} Gc^2 \\ \leq &\frac{1}{\sqrt{T}} D_f + \frac{1}{2\sqrt{T}} Gc^2 \end{aligned} \tag{15}$$

where $D_f = f(x_1) - \min_x f(x)$. $\qquad\square$

# B  Proof of Theorem 2

In the proof, we assume $\xi_t \sim \tilde{p}_t$ we omit subscript of $\mathbb{P}$ and $\mathbb{E}$ to simplify notations.

## B.1  When gradient is small

Let us first consider the case with $\|\nabla f(x_t)\| \leq \frac{3}{4} c$.

Denote $B$ to be the event that $\|\nabla f(x_t) + \xi_t\| \leq c$ and $\|\nabla f(x_t) - \xi_t\| \leq c$, we have $\mathbb{P}(B) \geq \mathbb{P}(\|\xi_t\| \leq \frac{c}{4})$. Define $D = \{\xi : \|\nabla f(x_t) + \xi_t\| > c \text{ or } \|\nabla f(x_t) - \xi_t\| > c\}$. Taking an expectation conditioning on $x_t$, we have

$$\begin{aligned} &\mathbb{E}[\langle \nabla f(x_t), g_t \rangle] \\ =&\langle \nabla f(x_t), \mathbb{E}[\mathrm{clip}(\nabla f(x_t) + \xi_t, c)] \rangle \\ =&\left\langle \nabla f(x_t), \mathbb{E}\left[\mathrm{clip}(\nabla f(x_t) + \xi_t, c) \middle| B\right] \right\rangle \mathbb{P}(B) \\ &+ \left\langle \nabla f(x_t), \int_D \mathrm{clip}(\nabla f(x_t) + \xi_t, c)\tilde{p}(\xi_t)d\xi_t \right\rangle \\ \geq&\|\nabla f(x_t)\|^2 \mathbb{P}(\|\xi_t\| \leq \frac{c}{4}) + \underbrace{\left\langle \nabla f(x_t), \int_D \mathrm{clip}(\nabla f(x_t) + \xi_t, c)\tilde{p}(\xi_t)d\xi_t \right\rangle}_{T_1} \end{aligned}$$

where the last inequality is due to $\mathrm{clip}(\nabla f(x_t) + \xi_t, c) = \nabla f(x_t) + \xi_t$ when $B$ happens and $\mathbb{P}(B) \geq \mathbb{P}(\|\xi_t\| \leq \frac{c}{4})$ and $\tilde{p}(\xi) = \tilde{p}(-\xi)$.

Now we need to look at $T_1$.

We have

$$T_1 = \frac{1}{2}\left(\left\langle \nabla f(x_t), \int_D \text{clip}(\nabla f(x_t) + \xi_t, c)\tilde{p}(\xi_t)d\xi_t \right\rangle + \left\langle \nabla f(x_t), \int_D \text{clip}(\nabla f(x_t) - \xi_t, c)\tilde{p}(\xi_t)d\xi_t \right\rangle\right)$$

$$= \frac{1}{2}\left\langle \nabla f(x_t), \int_D \left(\text{clip}(\nabla f(x_t) + \xi_t, c) + \text{clip}(\nabla f(x_t) - \xi_t, c)\right)\tilde{p}(\xi_t)d\xi_t \right\rangle$$

$$= \frac{1}{2}\|\nabla f(x_t)\|$$

$$\times \int_D \underbrace{\left(\|\text{clip}(\bar{g}_t + \xi_t, c)\|cos(\bar{g}_t, \bar{g}_t + \xi_t) + \|\text{clip}(\bar{g}_t - \xi_t, c)\|cos(\bar{g}_t, \bar{g}_t - \xi_t)\right)}_{T_2(\xi_t)}\tilde{p}(\xi_t)d\xi_t \quad (16)$$

where $\bar{g}_t \triangleq \nabla f(x_t)$ and the last equality is because $\langle a, b \rangle = \|a\|\|b\| \cos(a, b)$ for any vector $a, b$, and that the clipping operation keeps directions.

Now it reduces to analyzing $T_2(\xi_t)$. Our target now is to prove $T_2(\xi_t) \geq 0$.

Let us first consider the case where $\|\bar{g}_t + \xi_t\| \geq c$ and $\|\bar{g}_t - \xi_t\| \geq c$. In this case, we have

$$T_2(\xi) = c(cos(\bar{g}_t, \bar{g}_t + \xi_t) + cos(\bar{g}_t, \bar{g}_t - \xi_t)) \geq 0 \quad (17)$$

where the inequality is due to Lemma 1.

Another case is one of $\|\bar{g}_t + \xi_t\|$ and $\|\bar{g}_t - \xi_t\|$ is less than $c$. Assume $\cos(\bar{g}_t, \bar{g}_t - \xi_t) < 0$. In this case, we must have $\cos(\bar{g}_t, -\xi_t) < 0$ and $cos(\bar{g}_t, \bar{g}_t + \xi_t) > 0$ from basic properties of trigonometric functions. Then, from Lemma 2, we have

$$\|\bar{g}_t + \xi\| \geq \|\bar{g}_t - \xi\|. \quad (18)$$

So in this case, we have

$$T_2(\xi_t) = \|\text{clip}(\bar{g}_t + \xi_t, c)\| \cos(\bar{g}_t, \bar{g}_t + \xi_t) + \|\text{clip}(\bar{g}_t - \xi, c)\|cos(\bar{g}_t, \bar{g}_t - \xi)$$

$$= c \cdot \cos(\bar{g}_t, \bar{g}_t + \xi_t) + \|\text{clip}(\bar{g}_t - \xi_t, c)\| \cos(\bar{g}_t, \bar{g}_t - \xi_t)$$

$$\geq c \cdot \cos(\bar{g}_t, \bar{g}_t + \xi_t) + c \cdot cos(\bar{g}_t, \bar{g}_t - \xi_t)$$

$$\geq 0 \quad (19)$$

where the last inequality is due to Lemma 1.

Similar argument applies to the case with $cos(\bar{g}_t, \bar{g}_t + \xi_t) < 0$.

For the case with $cos(\bar{g}_t, \bar{g}_t + \xi_t) \geq 0$ and $cos(\bar{g}_t, \bar{g}_t - \xi_t) \geq 0$, we trivially have $T_2(\xi_t) \geq 0$. Thus, we have

$$\mathbb{E}[\langle \nabla f(x_t), g_t \rangle] \geq \|\nabla f(x_t)\|^2 \mathbb{P}(\|\xi_t\| \leq \frac{c}{4}). \quad (20)$$

This completes the proof. $\qquad\square$

## B.2 When gradient is large

Now let us look at the case where gradient is large, i.e. $\|\nabla f(x_t)\| \geq \frac{3}{4}c$.

By definition, we have

$$\mathbb{E}[\langle \nabla f(x_t), g_t \rangle]$$

$$= \left\langle \nabla f(x_t), \int_\xi \text{clip}(\nabla f(x_t) + \xi, c)p(\xi)d\xi \right\rangle$$

$$= \int_\xi \langle \nabla f(x_t), \text{clip}(\nabla f(x_t) + \xi, c)p(\xi)d\xi \rangle$$

$$= \|\nabla f(x_t)\| \underbrace{\int_\xi \|\text{clip}(\nabla f(x_t) + \xi, c)\|cos(\nabla f(x_t), \nabla f(x_t) + \xi)p(\xi)d\xi}_{T_7} \quad (21)$$

where the last equality is by definition of cosine and the fact that the clipping operation keeps directions.

In the following, we want to show $T_7$ is an non-decreasing function of $\|\nabla f(x_t)\|$, then the result can be directly obtained from first part of the theorem.

Notice that $T_7$ is invariant to simultaneous rotation of $\nabla f(x_t)$ and the noise distribution (i.e., changing the coordinate axis of the space). Thus, wlog, we can assume $\nabla f(x_t)_1 = y > 0$ and $\nabla f(x_t)_i = 0$ for $2 \leq i \leq d$. To show $T_7$ is a non-decreasing function of $\|\nabla f(x_t)\|$, it is enough to show each term in the integration is an non-decreasing function of $y$. I.e., it is enough to show that, for all $\xi_t$, the following quantity

$$\|\text{clip}(\nabla f(x_t) + \xi_t, c)\| cos(\nabla f(x_t), \nabla f(x_t) + \xi_t) \tag{22}$$

is an non-decreasing function of $y$ for $y > 0$ when $\nabla f(x_t) = [y, 0, ..., 0]$.

First consider the case where $\|\nabla f(x_t) + \xi_t\| \leq c$. In this case, (22) reduces to

$$\begin{aligned}
&\|\nabla f(x_t) + \xi_t\| cos(\nabla f(x_t), \nabla f(x_t) + \xi_t)\\
=&\|\nabla f(x_t) + \xi_t\| \frac{\langle \nabla f(x_t), \nabla f(x_t) + \xi_t \rangle}{\|\nabla f(x_t)\|\|\nabla f(x_t) + \xi_t\|}\\
=&\frac{\langle \nabla f(x_t), \nabla f(x_t) + \xi_t \rangle}{\|\nabla f(x_t)\|}\\
=&\frac{y(y + \xi_{t,1})}{y} = y + \xi_{t,1}
\end{aligned} \tag{23}$$

which is a monotonically increasing function of $y$.

Now consider the case with $\|\nabla f(x_t) + \xi_t\| \geq c$, we have

$$\begin{aligned}
&\|\text{clip}(\nabla f(x_t) + \xi, c)\| cos(\nabla f(x_t), \nabla f(x_t) + \xi_t)\\
=&c \cdot cos(\nabla f(x_t), \nabla f(x_t) + \xi_t)\\
=&c\frac{\langle \nabla f(x_t), \nabla f(x_t) + \xi_t \rangle}{\|\nabla f(x_t)\|\|\nabla f(x_t) + \xi_t\|}\\
=&c\frac{y(y + \xi_{t,1})}{y\sqrt{(y + \xi_{t,1})^2 + \sum_{i=2}^d \xi_{t,i}^2}} = c\frac{(y + \xi_{t,1})}{\sqrt{(y + \xi_{t,1})^2 + \sum_{i=2}^d \xi_{t,i}^2}}
\end{aligned} \tag{24}$$

which is a non-decreasing function of $y$.

To see it is non-decreasing, define

$$r(z) = c\frac{z}{\sqrt{z^2 + q^2}}, \tag{25}$$

we have $r'(z) = c(1 - \frac{z^2}{z^2+q^2}) \geq 0$. The term in RHS of (24) can be treated as $z = y + \xi_{t,1}$ and $q^2 = \sum_{i=2}^d \xi_{t,i}^2$.

Since the clipping function is continuous, combined with the above results, we know (22) is an non-decreasing function of $\|\nabla f(x_t)\|$.

Then we have

$$\begin{aligned}
&\mathbb{E}[\langle \nabla f(x_t), g_t \rangle]\\
=&\|\nabla f(x_t)\| \int_{\xi_t} \|\text{clip}(\nabla f(x_t) + \xi_t, c)\| cos(\nabla f(x_t), \nabla f(x_t) + \xi_t) p(\xi_t) d\xi_t\\
\geq&\|\nabla f(x_t)\| \int_{\xi_t} \|\text{clip}(\frac{3}{4}c\frac{\nabla f(x_t)}{\|\nabla f(x_t)\|} + \xi_t, c)\| cos(\frac{3}{4}c\frac{\nabla f(x_t)}{\|\nabla f(x_t)\|}, \frac{3}{4}c\frac{\nabla f(x_t)}{\|\nabla f(x_t)\|} + \xi_t) p(\xi_t) d\xi_t
\end{aligned} \tag{26}$$

From first part of the theorem, we know when $\|\nabla f(x_t)\| = \frac{3}{4}c$, we have

$$\mathbb{E}[\langle \nabla f(x_t), g_t \rangle] \geq \|\nabla f(x_t)\|^2 \mathbb{P}(\|\xi_t\| < \frac{c}{4}) = \|\nabla f(x_t)\| \left( \frac{3}{4}c \cdot \mathbb{P}(\|\xi_t\| < \frac{c}{4}) \right) \tag{27}$$

Combine the above result with (21) and the non-decreasing property of $T_7$, we see that when $\|\nabla f(x_t)\| \geq \frac{3}{4}c$, the following holds:

$$\|\nabla f(x_t)\| \left(\frac{3}{4}c \cdot \mathbb{P}(\|\xi\| < \frac{c}{4})\right) \leq \mathbb{E}[\langle \nabla f(x_t), g_t \rangle] = \|\nabla f(x_t)\| T_7,$$

which implies $T_7 \geq \frac{3}{4}c \cdot \mathbb{P}(\|\xi\| < \frac{c}{4})$. Substituting this lower bound into (21) finishes the proof. $\square$

### B.3 Technical lemmas

**Lemma 1.** *For any $g$ and $\xi$, we have*

$$cos(g, g+\xi) + cos(g, g-\xi) \geq 0 \tag{28}$$

**Proof:** By definition of $cos$, we have

$$
\begin{aligned}
&cos(g, g+\xi) + cos(g, g-\xi) \\
=&\frac{\langle g, g+\xi \rangle}{\|g\|\|g+\xi\|} + \frac{\langle g, g-\xi \rangle}{\|g\|\|g-\xi\|} \\
=&\frac{\|g\|}{\|g+\xi\|} + \frac{\|g\|}{\|g-\xi\|} + \frac{\langle g, \xi \rangle}{\|g\|\|g+\xi\|} - \frac{\langle g, \xi \rangle}{\|g-\xi\|} \\
=&\frac{\|g\|}{\|g+\xi\|} + \frac{\|g\|}{\|g-\xi\|} + \frac{\|\xi\|cos(g,\xi)}{\|g+\xi\|} - \frac{\|\xi\|cos(g,\xi)}{\|g-\xi\|} \\
=&\frac{\|g+\xi\|(\|g\|-\|\xi\|e) + \|g-\xi\|(\|g\|+\|\xi\|e)}{\|g+\xi\|\|g-\xi\|} 
\end{aligned}
\tag{29}
$$

where $e = cos(g, \xi)$.

To prove the desired result, we only need the numerator of RHS of (29) to be non-negative.

Denote $h(\xi) = cos(g, g+\xi) + cos(g, g-\xi)$, since $h$ is rotation invariant, we can assume $\xi_1 > 0$ and $\xi_{t,i} = 0$ for $2 \leq i \leq d$ wlog. Also, because $h(\xi) = h(-\xi)$, we can assume $g_1 \geq 0$ wlog.

Now suppose $g_1 = a$, $\sum_{i=2}^{d} g_i^2 = b^2$, Denote the numerator of RHS of (29) as $T_3$, it can be written as

$$
\begin{aligned}
T_3 =&\|g+\xi\|(\|g\|-\|\xi\|e) + \|g-\xi\|(\|g\|+\|\xi\|e) \\
=&\underbrace{\sqrt{b^2 + (a+\xi_1)^2}(\sqrt{a^2+b^2} - \xi_1 e)}_{T_4} + \underbrace{\sqrt{b^2 + (a-\xi_1)^2}(\sqrt{a^2+b^2} + \xi_1 e)}_{T_5}
\end{aligned}
\tag{30}
$$

and $e = \frac{\langle g, \xi \rangle}{\|g\|\|\xi\|} = \frac{a}{\sqrt{a^2+b^2}}$.

Now let us analyze when $T_3$ can be possibly less than 0. Recall that by assumption, $\xi_1 > 0$ and $e \geq 0$. Then we know $T_5 \geq 0$. We have $T_3 \geq 0$ trivially when $T_4 \geq 0$, i.e. when $\xi_1 e \leq \sqrt{a^2+b^2}$.

Now assume $\xi_1 e > \sqrt{a^2+b^2}$. To ensure $T_3 \geq 0$, we can alternatively ensure $T_5^2 \geq T_4^2$ in this case.

We have

$$
\begin{aligned}
T_5^2 - T_4^2 =&(b^2 + (a-\xi_1)^2)(\sqrt{a^2+b^2} + \xi_1 e)^2 - (b^2 + (a+\xi_1)^2)(\sqrt{a^2+b^2} - \xi_1 e)^2 \\
=&4b^2 \xi_1 e\sqrt{a^2+b^2} + \underbrace{4\xi_1 e\sqrt{a^2+b^2}(a^2 + \xi_1^2) - 4a\xi_1(a^2 + b^2 + \xi_1^2 e^2)}_{T_6}
\end{aligned}
\tag{31}
$$

For $T_6$, we can further simplify it as

$$
\begin{aligned}
T_6 =&4\xi_1 e\sqrt{a^2+b^2}(a^2 + \xi_1^2) - 4a\xi_1(a^2 + b^2 + \xi_1^2 e^2) \\
=&4\xi_1 a(a^2 + \xi_1^2) - 4a\xi_1(a^2 + b^2 + \xi_1^2 e^2) \\
=&4\xi_1 a(\xi_1^2(1 - e^2) - b^2) \\
=&4\xi_1 a(\xi_1^2(\frac{b^2}{a^2+b^2}) - b^2) \\
=&4\xi_1 a(b^2(\frac{\xi_1^2 - (a^2+b^2)}{a^2+b^2})) \\
\geq&0
\end{aligned}
\tag{32}
$$

where the last inequality is because $\xi^2 \geq \xi^2 e^2 \geq a^2 + b^2$ and $\xi_1 a > 0$ as assumed previously.

Combining all above, we have $T_6 \geq 0 \implies T_5^2 - T_4^2 \geq 0 \implies T_3 \geq 0 \implies cos(g, g + \xi) + cos(g, g - \xi) \geq 0$ which proves the desired result. $\square$

**Lemma 2.** *For any $g$ and $\xi$, we have*

$$\|g + \xi\| \geq \|g - \xi\| \tag{33}$$

*if $cos(g, \xi) > 0$ and*

$$\|g + \xi\| \leq \|g - \xi\| \tag{34}$$

*if $cos(g, \xi) < 0$.*

**Proof:** Express $\xi$ using a coordinate system with one axis parallel to $g$. Define the basis of this coordinate system as $v_1, v_2, ... v_d$ with $v_1 = g/\|g\|$. Then we have $\xi = \sum_{i=1}^{d} b_i v_i$ and $cos(g, \xi) > 0$ if and only if $b_1 > 0$.

In addition, we have

$$\|g + \xi\| = \sqrt{(\|g\| + b_1)^2 + \sum_{i=2}^{d} b_i^2} \tag{35}$$

and

$$\|g - \xi\| = \sqrt{(\|g\| - b_1)^2 + \sum_{i=2}^{d} b_i^2}. \tag{36}$$

Then it is clear that $\|g + \xi\| \geq \|g - \xi\|$ when $b_1 > 0$ which means $cos(g, \xi) > 0$.

Similar arguments applies to the case with $cos(g, \xi) < 0$ $\square$

## C  Proof of Theorem 3

**Theorem 3.** *Given $m$ distributions with the pdf of the $i$th distribution being $p_i(\xi_t) = \phi_i(\|\xi_t - u_i\|)$ for some function $\phi_i$. If $\nabla f(x_t) = \sum_{i=1}^{m} w_i u_i$ for some $w_i \geq 0, \sum_{i=1}^{m} w_i = 1$. Let $p'(\xi_t) = \sum_{i=1}^{m} w_i p_i(\xi_t - \nabla f(x_t))$, be a mixture of these distributions with zero mean. If $\langle u_i, \nabla f(x_t) \rangle \geq 0, \forall i \in [m]$, we have*

$$\mathbb{E}_{\xi_t \sim p'}[\langle \nabla f(x_t), g_t \rangle] \geq \|\nabla f(x_t)\| \sum_{i=1}^{m} w_i \min(\|u_i\|, \frac{3}{4}c) \cos(\nabla f(x_t), u_i) \mathbb{P}_{\xi_t \sim p_i}(\|\xi_t\| < \frac{c}{4}) \geq 0$$

**Proof:** First, we notice that Theorem 2 can be restated into a more general form as follows.

**Theorem 2** (restated). *Given a random variable $\xi \sim \tilde{p}$ with $\tilde{p}(\xi) = \tilde{p}(-\xi)$ being a symmetric distribution, for any vector $g$, we have*

$$\text{1. If } \|g\| \leq \frac{3}{4}c, \quad \text{then} \quad \mathbb{E}[\langle g, clip(g + \xi, c) \rangle] \geq \|g\|^2 \mathbb{P}\left(\|\xi\| < \frac{c}{4}\right) \tag{37}$$

$$\text{2. If } \|g\| > \frac{3}{4}c, \quad \text{then} \quad \mathbb{E}[\langle g, clip(g + \xi, c) \rangle] \geq \frac{3}{4}c\|g\|\mathbb{P}\left(\|\xi\| < \frac{c}{4}\right) \tag{38}$$

In addition, by sphere symmetricity, if $\xi \sim \hat{p}$ with $\hat{p}$ being a spherical distribution $\hat{p}(\xi) = \phi(\|\xi\|)$ for some function $\phi$, for any vector $g$, we have $\mathbb{E}[clip(g + \xi)] = rg$ with $r$ being a constant (i.e. the expected clipped gradient is in the same direction as $g$). Combining with restated Theorem 2 above, we have when $\tilde{p}$ is a spherical distribution with $\tilde{p}(\xi) = \phi(\|\xi\|)$,

$$\mathbb{E}[clip(g + \xi, c)] = rg \tag{39}$$

with $r \geq 0$ and

$$r\|g\| \geq \min(\frac{3}{4}c, \|g\|)\mathbb{P}\left(\|\xi\| < \frac{c}{4}\right). \tag{40}$$

Now we can use the above results to prove the theorem.

The expectation can be splitted as

$$\mathbb{E}_{\xi_t \sim p'}[\langle \nabla f(x_t), g_t \rangle] = \sum_{i=1}^{m} w_i \mathbb{E}_{\xi_t \sim p_i}[\langle \nabla f(x_t), g_t \rangle]. \tag{41}$$

Then, because (39) and $E_{\xi_t \sim p_i}[g_t] = u_i$ and that $p_i$ corresponds to a noise with spherical distribution added to $u_i$, we have

$$E_{\xi_t \sim p_i}[\langle \nabla f(x_t), g_t \rangle] = \langle \nabla f(x_t), E_{\xi_t \sim p_i}[g_t] \rangle = \langle \nabla f(x_t), r_i u_i \rangle \tag{42}$$

with $r_i \|u_i\| \geq \min(\frac{3}{4}c, \|u_i\|)\mathbb{P}_{\xi_t \sim p_i}\left(\|\xi_t\| < \frac{c}{4}\right)$. Since we assumed $\langle u_i, \nabla f(x_t) \rangle \geq 0$, we have

$$\mathbb{E}_{\xi_t \sim p'}[\langle \nabla f(x_t), g_t \rangle] \geq \|\nabla f(x_t)\| \sum_{i=1}^{m} w_i \min(\frac{3}{4}c, \|u_i\|)\cos(u_i, \nabla f(x_t))\mathbb{P}_{\xi_t \sim p_i}\left(\|\xi_t\| < \frac{c}{4}\right) \geq 0 \tag{43}$$

which is the desired result. $\qquad\square$

# D  Proof of Theorem 5

Recall the algorithm has the following update rule

$$x_{t+1} = x_t - \alpha\left(\left(\frac{1}{|S_t|}\sum_{i \in S_t}\text{clip}(\nabla f(x_t) + \xi_{t,i}, c)\right) + Z_t\right) \tag{44}$$

where $g_{t,i} \triangleq \nabla f(x_t) + \xi_{t,i}$ is the stochastic gradient at iteration $t$ evaluated on sample $i$, and $S_t$ is a subset of whole dataset $D$; $Z_t \sim \mathcal{N}(0, \sigma^2 I)$ is the noise added for privacy. We denote $g_t = \frac{1}{|S_t|}\sum_{i \in S_t}\text{clip}(\nabla f(x_t) + \xi_{t,i}, c)$ in the remaining parts of the proof to simplify notation.

Following traditional convergence analysis of SGD using smoothness assumption, we first have

$$f(x_{t+1}) \leq f(x_t) + \langle \nabla f(x_t), x_{t+1} - x_t \rangle + \frac{1}{2}G\|x_{t+1} - x_t\|^2 \tag{45}$$

which translates into

$$f(x_{t+1}) \leq f(x_t) - \alpha\langle \nabla f(x_t), (g_t + Z_t) \rangle + \frac{1}{2}G\alpha^2\|g_t + Z_t\|^2 \tag{46}$$

Taking expectation conditioned on $x_t$, we have

$$\mathbb{E}[f(x_{t+1})]$$
$$\leq f(x_t) - \alpha\mathbb{E}[\langle \nabla f(x_t), g_t \rangle] + \frac{1}{2}G\alpha^2(\mathbb{E}[\|g_t\|^2] + \sigma^2 c^2 d)$$
$$\leq f(x_t) - \alpha\mathbb{E}[\langle \nabla f(x_t), g_t \rangle] + \frac{1}{2}G\alpha^2(c^2 + \sigma^2 c^2 d). \tag{47}$$

Take overall expectation and sum over $t \in [T]$ and rearrange, we have

$$\sum_{t=1}^{T}\alpha\mathbb{E}[\langle \nabla f(x_t), g_t \rangle] \leq f(x_1) - \mathbb{E}[f(x_{T+1})] + T\frac{1}{2}G\alpha^2(c^2 + \sigma^2 d). \tag{48}$$

Dividing both sides by $T\alpha$, we get

$$\frac{1}{T}\sum_{t=1}^{T}\mathbb{E}[\langle \nabla f(x_t), g_t \rangle] \leq \frac{f(x_1) - \mathbb{E}[f(x_{T+1})]]}{T\alpha} + \frac{1}{2}G\alpha(c^2 + \sigma^2 d). \tag{49}$$

To achieve $(\epsilon, \delta)$-privacy, we need $\sigma^2 = v\frac{Tc^2\ln(\frac{1}{\delta})}{n^2\epsilon^2}$ for some constant $v$ by Theorem 1 in Abadi et al. [2016b]. Substituting the expression of $\sigma^2$ into the above inequality, we get

$$\frac{1}{T}\sum_{t=1}^{T}\langle\mathbb{E}[\nabla f(x_t), g_t]\rangle \leq \frac{D_f}{T\alpha} + \frac{1}{2}G\alpha(c^2 + v\frac{T\ln(\frac{1}{\delta})}{n^2\epsilon^2}c^2d) \tag{50}$$

where we define $D_f = f(x_1) - \min_x f(x)$.

Setting $T\alpha = \frac{\sqrt{D_f}n\epsilon}{\sqrt{G}c\sqrt{d}\sqrt{\ln(\frac{1}{\delta})}}$, we have

$$\frac{1}{T}\sum_{t=1}^{T}\mathbb{E}[\langle\nabla f(x_t), g_t\rangle] \leq \left(\frac{1}{2}v + 1\right)\frac{c\sqrt{D_f Gd\ln(\frac{1}{\delta})}}{n\epsilon} + \frac{1}{2}G\alpha c^2. \tag{51}$$

Setting $\alpha = \frac{\sqrt{D_f d\ln(\frac{1}{\delta})}}{n\epsilon c\sqrt{G}}$, we have

$$\frac{1}{T}\sum_{t=1}^{T}\mathbb{E}[\langle\nabla f(x_t), g_t\rangle] \leq \left(\frac{1}{2}v + \frac{3}{2}\right)\frac{c\sqrt{D_f Gd\ln(\frac{1}{\delta})}}{n\epsilon}. \tag{52}$$

The remaining step is to analyze the term on LHS of (52). We first notice that the gradient sampling scheme yields

$$\mathbb{E}[\langle\nabla f(x_t), g_t\rangle] = \mathbb{E}_{\xi_t\sim p}[\langle\nabla f(x_t), \text{clip}(\nabla f(x_t) + \xi_t, c)\rangle] \tag{53}$$

with $\xi_t$ being a discrete random variable that can takes values $\xi_{t,i}, i \in D$ with equal probability and $D$ is the whole dataset.

Now it is time to split the bias as following.

$$\mathbb{E}_{\xi_t\sim p}[\langle\nabla f(x_t), g_t\rangle] = \mathbb{E}_{\xi_t\sim\tilde{p}}[\langle\nabla f(x_t), g_t\rangle] + \int\langle\nabla f(x_t), \text{clip}(\nabla f(x_t) + \xi_t, c)\rangle(p_t(\xi_t) - \tilde{p}_t(\xi_t))d\xi_t$$

with $\tilde{p}$ being a symmetric distribution. Applying Theorem 2, we have

$$\mathbb{E}_{\xi_t\sim\tilde{p}}[\langle\nabla f(x_t), g_t,\rangle] \geq \mathbb{P}_{\xi_t\sim\tilde{p}}(\|\xi_t\| < \frac{c}{4})\|\nabla f(x_t)\|^2 \tag{54}$$

when $\|\nabla f(x_t)\| \leq \frac{3}{4}c$ and

$$\mathbb{E}_{\xi_t\sim\tilde{p}}[\langle\nabla f(x_t), g_t\rangle] \geq \frac{3}{4}\mathbb{P}_{\xi_t\sim\tilde{p}}(\|\xi_t\| < \frac{c}{4})c\|\nabla f(x_t)\| \tag{55}$$

when $\|\nabla f(x_t)\| \geq \frac{3}{4}c$.

Now we bound the bias term using Wasserstein distance as follows.

$$-\int\langle\nabla f(x_t), \text{clip}(\nabla f(x_t) + \xi_t, c)\rangle(p_t(\xi_t) - \tilde{p}_t(\xi_t))d\xi_t$$

$$= \int\langle\nabla f(x_t), \text{clip}(\nabla f(x_t) + \xi_t, c)\rangle(\tilde{p}(\xi_t) - p(\xi_t))d\xi_t$$

$$= \int\langle\nabla f(x_t), \text{clip}(\nabla f(x_t) + \xi_t, c)\rangle\tilde{p}(\xi_t)d\xi_t - \int\langle\nabla f(x_t), \text{clip}(\nabla f(x_t) + \xi_t', c)\rangle p(\xi_t')d\xi_t'$$

$$= \int\int(\langle\nabla f(x_t), \text{clip}(\nabla f(x_t) + \xi_t, c)\rangle - \langle\nabla f(x_t), \text{clip}(\nabla f(x_t) + \xi_t', c)\rangle)\gamma(\xi_t, \xi_t')d\xi_t d\xi_t'$$

$$\leq \int\int|\langle\nabla f(x_t), \text{clip}(\nabla f(x_t) + \xi_t, c)\rangle - \langle\nabla f(x_t), \text{clip}(\nabla f(x_t) + \xi_t', c)\rangle|\gamma(\xi_t, \xi_t')d\xi_t d\xi_t' \tag{56}$$

where $\gamma$ is any joint distribution with marginal $\tilde{p}$ and $p$. Thus, we have

$$-\int\langle\nabla f(x_t), \text{clip}(\nabla f(x_t) + \xi_t, c)\rangle(p_t(\xi_t) - \tilde{p}_t(\xi_t))d\xi_t$$

$$\leq \inf_{\gamma\in\Gamma(\tilde{p},p)}\int\int|\langle\nabla f(x_t), \text{clip}(\nabla f(x_t) + \xi_t, c)\rangle - \langle\nabla f(x_t), \text{clip}(\nabla f(x_t) + \xi_t', c)\rangle|\gamma(\xi_t, \xi_t')d\xi_t d\xi_t'$$

where $\Gamma(\tilde{p}, p)$ is the set of all coupling with marginals $\tilde{p}$ and $p$ on the two factors, respectively. If we define the distance function $d_{y,c}(a, b) = |\langle y, \text{clip}(y + a, c) \rangle - \langle y, \text{clip}(y + b, c) \rangle|$, we have

$$- \int \langle \nabla f(x_t), \text{clip}(\nabla f(x_t) + \xi_t, c) \rangle (p_t(\xi_t) - \tilde{p}_t(\xi_t)) d\xi_t$$

$$\leq \inf_{\gamma \in \Gamma(\tilde{p}, p)} \int \int d_{\nabla f(x_t), c}(\xi_t, \xi_t') \gamma(\xi_t, \xi_t') d\xi_t d\xi_t' = W_{\nabla f(x_t), c}(\tilde{p}_t, p_t) \qquad (57)$$

which we define $W_{v,c}(p, p')$ as the Wasserstein distance between $p$ and $p'$ using the metric $d_{v,c}$.

Wrapping up, define

$$h(y) = \begin{cases} y^2, & \text{for } y \leq 3c/4 \\ \frac{3}{4}cy, & \text{for } y > 3c/4 \end{cases},$$

we have

$$\frac{1}{T} \sum_{t=1}^{T} \mathbb{P}_{\xi_t \sim \tilde{p}_t}(\|\xi_t\| < \frac{c}{4}) h(\|\nabla f(x_t)\|) \leq \left(\frac{1}{2}v + \frac{3}{2}\right) \frac{c\sqrt{D_f G d \ln(\frac{1}{\delta})}}{n\epsilon} + \frac{1}{T} \sum_{t=1}^{T} W_{\nabla f(x_t), c}(\tilde{p}_t, p_t)$$

$$(58)$$

which is the desired result. $\qquad\square$

# E    Proof of Theorem 6

**Theorem 6.** *Let $g_t = clip(\nabla f(x_t) + \xi_t + k\zeta_t, c)$ and $\zeta_t \sim \mathcal{N}(0, I)$. Then gradient clipping algorithm has following properties:*

$$\mathbb{E}_{\xi_t \sim p, \zeta_t}[\langle \nabla f(x_t), g_t \rangle] \geq \|\nabla f(x_t)\| \min \left\{ \|\nabla f(x_t)\|, \frac{3}{4}c \right\} \mathbb{P}(\|k\zeta_t\| < \frac{c}{4}) - O(\frac{\sigma_{\xi_t}^2}{k^2}) \qquad (59)$$

*where $\sigma_{\xi_t}^2$ is the variance of the gradient noise $\xi_t$.*

**Proof**: Define $W_t = \xi_t + k\zeta_t$ be the total noise on the gradient before clipping and $W_t \sim \bar{p}$. We know $\mathbb{E}[W_t] = 0$ and $\bar{p}(W_t) = \int_{\xi_t} p(\xi_t) \frac{1}{k} \psi(\frac{W_t - \xi_t}{k}) d\xi_t$ with $\psi$ being the pdf of $\mathcal{N}(0, I)$. The proof idea is to bound the total variation distance between $\bar{p}(W_t)$ and $\frac{1}{k}\psi$ as $O(\frac{\sigma_{\xi_t}^2}{k^2})$, then use this distance to bound the clipping bias $b_t$. This implies $\bar{p}(W_t)$ will become more and more symmetric as $k$ increases.

We have

$$\int \left| \bar{p}(W_t) - \frac{1}{k}\psi(\frac{W_t}{k}) \right| dW_t$$

$$= \int_{W_t} \left| \int_{\xi_t} p(\xi_t) \frac{1}{k} \psi(\frac{W_t - \xi_t}{k}) d\xi_t - \frac{1}{k}\psi(\frac{W_t}{k}) \right| dW_t$$

$$= k \int_{W_t'} \left| \int_{\xi_t} p(\xi_t) \frac{1}{k} \psi(W_t' - \frac{\xi_t}{k}) d\xi_t - \frac{1}{k}\psi(W_t') \right| dW_t' \qquad (60)$$

By Taylor's series, we have

$$\psi(W_t' - \frac{\xi_t}{k}) = \psi(W_t') + \langle \nabla \psi(W_t'), \frac{-\xi_t}{k} \rangle + \int_0^1 \left\langle \frac{\xi_t}{k}, \nabla^2 \psi(W_t' - \tau\frac{\xi_t}{k})\frac{\xi_t}{k} \right\rangle (1 - \tau) d\tau \qquad (61)$$

Then,

$$\int |\bar{p}(W_t) - \frac{1}{k}\psi(\frac{W_t}{k})| dW_t$$

$$= \int_{W_t'} \left| \int_{\xi_t} p(\xi_t) \psi(W_t' - \frac{\xi_t}{k}) d\xi_t - \psi(W_t') \right| dW_t'$$

$$= \int_{W_t'} \left| \int_{\xi_t} p(\xi_t) \int_0^1 \langle \frac{\xi_t}{k}, \nabla^2 \psi(W_t' - t\frac{\xi_t}{k})\frac{\xi_t}{k} \rangle (1 - \tau) dt d\xi_t \right| dW_t'$$

$$\leq \int_0^1 \int_{\xi_t} \int_{W_t'} \left| p(\xi_t) \langle \frac{\xi_t}{k}, \nabla^2 \psi(W_t' - t\frac{\xi_t}{k})\frac{\xi_t}{k} \rangle (1 - \tau) \right| dW_t' d\xi_t d\tau \qquad (62)$$

where the second equality is obtained by applying (61) and using the fact that $\xi_t$ is zero mean.

Noticing that $\tau \leq 1$ and define $\hat{W}_t = W'_t - \tau \frac{\xi_t}{k}$, we have

$$\int_{W'_t} \left| p(\xi_t) \langle \frac{\xi_t}{k}, \nabla^2 \psi(W'_t - \tau \frac{\xi_t}{k}) \frac{\xi_t}{k} \rangle (1 - \tau) \right| dW'_t$$

$$= p(\xi_t)(1 - \tau) \int_{\hat{W}_t} \left| \langle \frac{\xi_t}{k}, \nabla^2 \psi(\hat{W}'_t) \frac{\xi_t}{k} \rangle \right| \frac{dW'_t}{d\hat{W}_t} d\hat{W}_t$$

$$= p(\xi_t)(1 - \tau) \int_{\hat{W}_t} \left| \langle \frac{\xi_t}{k}, \nabla^2 \psi(\hat{W}_t) \frac{\xi_t}{k} \rangle \right| d\hat{W}_t \qquad (63)$$

and the integration term only depends on $\|\frac{\xi_t}{k}\|$ due to sphere symmetricity of $\psi$. Thus we can assume $\xi_{t,1} = \|\xi_t\|$ and $\xi_{t,i} = 0$ for $i \geq 2$, wlog. Then, we have

$$\int_{W'_t} \left| p(\xi_t) \langle \frac{\xi_t}{k}, \nabla^2 \psi(W'_t - \tau \frac{\xi_t}{k}) \frac{\xi_t}{k} \rangle (1 - \tau) \right| dW'_t$$

$$\leq p(\xi_t)(1 - \tau) \int_{W'_t} \frac{\|\xi_t\|^2}{k^2} \left| \nabla^2_{1,1} \psi(W'_t - \tau \frac{\xi_t}{k}) \right| dW'_t$$

$$\leq p(\xi_t)(1 - \tau) \int_{\hat{W}_t} \frac{\|\xi_t\|^2}{k^2} \left| \nabla^2_{1,1} \psi(W_t) \right| \frac{dW'_t}{d\hat{W}_t} d\hat{W}_t$$

$$\leq p(\xi_t)(1 - \tau) \frac{\|\xi_t\|^2}{k^2} q \qquad (64)$$

where we have define $\hat{W}_t = W'_t - \tau \frac{\xi_t}{k}$ and $q = \int_{-\infty}^{\infty} |h''(x)| dx$ with $h(x)$ being the pdf of 1-dimensional standard normal distribution. Thus, $q$ is a dimension independent constant.

Substituting (64) into (62), we get

$$\int |\bar{p}(W_t) - \frac{1}{k} \psi(\frac{W_t}{k})| dW_t$$

$$\leq \int_0^1 \int_{\xi_t} \int_{W'_t} \left| p(\xi_t) \langle \frac{\xi_t}{k}, \nabla^2 \psi(W'_t - \tau \frac{\xi_t}{k}) \frac{\xi_t}{k} \rangle (1 - \tau) \right| dW'_t d\xi_t d\tau$$

$$\leq \int_0^1 \int_{\xi_t} p(\xi_t)(1 - \tau) \frac{\|\xi_t\|^2}{k^2} q d\xi_t d\tau$$

$$= \int_0^1 (1 - \tau) \frac{\sigma^2_{\xi_t}}{k^2} q d\tau$$

$$= \frac{1}{2} \frac{\sigma^2_{\xi_t}}{k^2} q \qquad (65)$$

where we used the fact that $\mathbb{E}[\xi_t] = 0$ and defined $\sigma^2_{\xi_t}$ being the variance of $\xi_t$.

By (5), we know

$$\mathbb{E}_{\xi_t \sim p, \zeta_t}[\langle \nabla f(x_t), g_t \rangle] = \mathbb{E}_{W_t \sim \tilde{p}}[\langle \nabla f(x_t), g_t \rangle]$$

$$+ \underbrace{\int \langle \nabla f(x_t), \text{clip}(\nabla f(x_t) + W_t, c) \rangle (p_t(W_t) - \tilde{p}_t(W_t)) dW_t}_{b_t} \qquad (66)$$

Let $\tilde{p}$ be the pdf of $k\zeta_t$, from Theorem 2, we have

$$\mathbb{E}_{W_t \sim \tilde{p}}[\langle \nabla f(x_t), g_t \rangle] \geq \|\nabla f(x_t)\| \min \left\{ \frac{3}{4} c, \|\nabla f(x_t)\| \right\} \mathbb{P}(\|k\zeta_t\| \leq \frac{c}{4}) \qquad (67)$$

In addition, we can bound $b_t$ as

$$|b_t| \leq \|\nabla f(x_t)\| c \int |p_t(\xi_t) - \tilde{p}_t(\xi_t)| d\xi_t \leq \|\nabla f(x_t)\| \frac{c}{2} \frac{\sigma^2_{\xi_t}}{k^2} q = O(\frac{\sigma^2_{\xi_t}}{k^2}) \qquad (68)$$

by (65).

Combining (66), (68), and (67) finishes the proof. $\qquad\square$

# F  More experiments on random projection

We show the projection of stochastic gradients into 2d space described in Section 4 for different projection matrices in Figure 5-8. It can be seen that as the training progresses, the gradient distribution in 2d space tend to be increasingly more symmetric.

(a) Repeat 1 　　　　(b) Repeat 2 　　　　(c) Repeat 3 　　　　(d) Repeat 4

(e) Repeat 5 　　　　(f) Repeat 6 　　　　(g) Repeat 7 　　　　(h) Repeat 8

Figure 5: Distribution of gradients on MNIST after epochs 0 projected using different random matrices.

(a) Repeat 1 　　　　(b) Repeat 2 　　　　(c) Repeat 3 　　　　(d) Repeat 4

(e) Repeat 5 　　　　(f) Repeat 6 　　　　(g) Repeat 7 　　　　(h) Repeat 8

Figure 6: Distribution of gradients on MNIST after epochs 3 projected using different random matrices.

(a) Repeat 1      (b) Repeat 2      (c) Repeat 3      (d) Repeat 4

(e) Repeat 5      (f) Repeat 6      (g) Repeat 7      (h) Repeat 8

Figure 7: Distribution of gradients on MNIST after epochs 9 projected using different random matrices.

(a) Repeat 1      (b) Repeat 2      (c) Repeat 3      (d) Repeat 4

(e) Repeat 5      (f) Repeat 6      (g) Repeat 7      (h) Repeat 8

Figure 8: Distribution of gradients on MNIST after epochs 59 projected using different random matrices.

# G  Evaluation on the probability term

In this section, we evaluate the probability term in Corollary 1 using a few statistics of the empirical gradient distribution on MNIST. Specifically, at the end of different epochs, we plot histogram of norm of stochastic gradient and norm of noise, along with the inner product between stochastic gradient (and clipped stochastic gradient) and the true gradient. The results are shown in Figure 9-11. One observation is that the norm of stochastic gradients is concentrated around 0 while having a heavy tail. The noise distribution is concentrated around some positive value with a heavy tail, the mode of the noise actually corresponds to the approximate 0 norm mode of stochastic gradients. As the training progress, the norm of stochastic gradients and the norm of noise are approaching 0. We set clipping threshold to be 1 in the experiment, so actually the probability $\mathbb{P}(\|\xi_t\| \leq \frac{1}{4}c)$ is 0 for the empirical distribution $p$. When we use a distribution $\tilde{p}$ with $\mathbb{P}(\|\xi_t\| \leq \frac{1}{4}c) \geq l$ for some value $l > 0$ to approximate $p$, this approximation indeed can create a approximation bias. However, the bias may not be too large since the mode of the norm of noise is not too much bigger than $\frac{c}{4}$. Furthermore, in Corollary 1 and Theorem 2, we actually can change $\mathbb{P}_{\xi_t \sim \tilde{p}}(\|\xi_t\| \leq \frac{1}{4}c)$ to $\mathbb{P}_{\xi_t \sim \tilde{p}}(\|\xi_t\| \leq zc)$ with any $z < 1$ and simultaneously change the $\frac{3}{4}c$ to $(1-z)c$ to make the probability term larger.

Despite the discussions above, the distribution of norm of stochastic gradients and noise norm combined with the 2d visualization experiments implies the noise on gradient might follow a mixture of distributions with each component being approximate symmetric. Especially one component may correspond to a approximate 0 mean distribution of stochastic gradients. Intuitively this can be true since each class of data may corresponds to a few variations of stochastic gradients and the gradient noise is observed to be low rank in Li et al. [2020]. We have some discussions in Section 2.2 to explain how convergence can be achieved in the cases of symmetric distribution mixtures but it may

not be the complete explanation here. Further exploration of gradient distribution in practice is an important question and we leave it for future research.

(a) Norm of gradients

(b) Norm of noise

(c) Inner product between true gradient and clipped stochastic gradients

(d) Inner product between true gradient and stochastic gradients

Figure 9: Distribution of different statistics at epoch 3.

(a) Norm of gradients

(b) Norm of noise

(c) Inner product between true gradient and clipped stochastic gradients

(d) Inner product between true gradient and stochastic gradients

Figure 10: Distribution of different statistics at epoch 9.

(a) Norm of gradients

(b) Norm of noise

(c) Inner product between true gradient and clipped stochastic gradients

(d) Inner product between true gradient and stochastic gradients

Figure 11: Distribution of different statistics at epoch 59.

# H   Additional results and discussions on the probability term and the noise adding approach in Section 5

Theorem 6 says that after adding the Gaussian noise $k\zeta_t$ before clipping, the clipping bias can decrease. In the meantime, the expected decent also decreases because $\mathbb{P}(\|k\zeta_t\| < \frac{c}{4})$ decreases with $k$. To get a more clear understanding of the theorem, consider $d = 1$, then $\mathbb{P}(\|k\zeta_t\| < \frac{c}{4}) = \text{erf}(\frac{c}{4k})$ which decreases with an order of $O(\frac{1}{k})$. This rate is slower than the $O(\frac{1}{k^2})$ diminishing rate of the clipping bias. Thus, as $k$ becomes large, the clipping bias will be negligible compared with the expected descent. This will translate to a **slower** convergence rate with a **better** final gradient bound in convergence analysis. The key idea of adding $k\zeta_t$ before clipping is to "symmetrify" the overall gradient noise distribution. By adding the isotropic symmetric noise $k\zeta_t$, the distribution of the resulting gradient noise $W_t \triangleq \xi_t + k\zeta_t$ will become increasingly more symmetric as $k$ increases. In particular, the total variation distance between the distribution of $W_t$ and $k\zeta_t$ decreases at a rate of $O(\frac{1}{k^2})$ which can be further used to bound the clipping bias. Then, one can apply Theorem 2 to lower bound $E_{\xi_t=0,\zeta_t}[\langle \nabla f(x_t), g_t \rangle]$ by letting $\tilde{p}$ be the distribution of $k\zeta_t$. We believe the lower bounds in Theorem 6 can be further improved when $d > 1$, notice that $\mathbb{P}(\|k\zeta_t\| < \frac{c}{4})$ tends to decrease fast with $k$ when $d$ being large.

However, we observe $E_{\xi_t \sim p,\zeta_t}[\langle \nabla f(x_t), g_t \rangle]$ decreases with a rate of $O(1/d)$ and $O(1/k)$ in practice for fixed $\|\nabla f(x_t)\|$ and $\xi_t = 0$ (see Table 1 for $\|\nabla f(x_t)\| = 10$, the expectation $E_{\xi_t=0,\zeta_t}[\langle \nabla f(x_t), g_t \rangle]$ is evaluated over $10^5$ samples of $\zeta_t \sim \mathcal{N}(0, I)$). In addition, we found the lower bounds in Theorem 2 are tight up to a constant when $d = 1$. To verify the lower bounds, we considered a 1-dimensional example and choose a symmetric noise $\xi_t \sim \mathcal{N}(0, 1)$ and set $c = 1$. Then we compare $\mathbb{E}_{\xi_t \sim \tilde{p}}[\langle \nabla f(x_t), g_t \rangle]$ (estimated by averaging $10^5$ samples) with the lower bound in Theorem 2 for different $\|\nabla f(x_t)\|$ and the results are shown in Table 2. Similar result should also hold for $\tilde{p}(\xi_t)$ being a distribution on a 1 dimensional subspace. This implies the lower bound can only be improved by using more properties of isotropic distributions like $\mathcal{N}(0, I)$ or resorting to a more general form of the lower bounds. We found this to be non-trivial and decide to leave it for future research.

Table 1: Scalability of $E_{\xi_t=0,\zeta_t}[\langle \nabla f(x_t), g_t \rangle]$ w.r.t. $d$ and $k$

|            | $d = 1$ | $d = 10$ | $d = 100$ | $d = 1000$ | $d = 10000$ |
|------------|---------|----------|-----------|------------|-------------|
| $k = 1$    | 10      | 9.572    | 7.077     | 3.015      | 0.995       |
| $k = 10$   | 6.788   | 2.961    | 0.992     | 0.316      | 0.1         |
| $k = 100$  | 0.758   | 0.316    | 0.098     | 0.032      | 0.01        |
| $k = 1000$ | 0.084   | 0.019    | 0.011     | 0.003      | 0.001       |

The results below verify our lower bound.

Table 2: Verify the lower bounds in Theorem 2

| $\|\nabla f(x_t)\|$ | 0.05 | 0.1 | 1 | 2 | 10 | 100 |
|---------------------|------|-----|---|---|----|-----|
| $\mathbb{E}_{\xi_t \sim \mathcal{N}(0,1)}[\langle \nabla f(x_t), g_t \rangle]$ | 1.7e-4 | 6.6e-3 | 0.612 | 1.83 | 10 | 100 |
| lower bound | 4e-5 | 2e-3 | 0.148 | 0.3 | 1.48 | 14.8 |