[Reviews · NeurIPS 2020]

Review 1

Summary and Contributions: In this paper authors analyze the convergence conditions for popular DP-SGD method by studying the geometric properties of bias introduced from gradient clipping and subsampling. Authors show that when the distribution of subsampling noise is symmetric, and the optimization problem satisfies certain smoothness conditions, clipped SGD will converge to correct solution. Next authors extend the analysis to DP-SGD and show similar convergence guarantees, even if the optimization problem is non-Lipschitz. Finally, authors illustrate that the gradient distribution for a certain deep learning problem is indeed nearly symmetric.

Strengths: The DP-SGD is widely applied method in differentially private machine learning, and studying the convergence of it is very important. This paper connects two fundamental parts of the DP-SGD, the subsampling induced noise and the clipping threshold, to guarantee the convergence, first in non-DP setting and later for full DP-SGD. The clipping induced bias is quantified in terms of a coupling between the true gradient distribution and a "close" symmetric distribution. This coupling gives a disparity measure between the two distributions. This gives an easy way of analyzing the convergence: when the true gradient distribution is close to symmetric, the DP-SGD behaves well. Besided the main analysis of the geometrics of DP-SGD, authors propose a method to mitigate the clipping induced bias for a case when the gradient distribution is not symmetric. They show by experiments that this method indeed works in three (toy) problems. However, authors should better connect this solution to DP-SGD, otherwise it feels bit unconnected to the rest of the nice work.

Weaknesses: This paper mainly deals with theoretical convergence guarantees of DP-SGD. However, the rate of convergence is not discussed in the paper. For DP-SGD, the convergence rate is very important, since more iterations will cause the privacy guarantees to detoriate. I think this angle should at least be mentioned somewhere in the discussion.

Correctness: I didn't go through all the proofs in supplementary, but the ones I read looked correct.

Clarity: The paper is well written. Typos: Line 125, Symetricity Line 132, sqaured Line 157, missing space after period ".Note"

Relation to Prior Work: Yes

Reproducibility: Yes

Additional Feedback: Equation 2, I think the \min is not supposed to be there. Or at least it should be on both sides. Equation 5, it's bit difficult to see how \xi affects the expression inside the expectation. Maybe you could remind that it is the noise that oracle "adds" to the gradient. Line 134, is there something missing from the sentence? It seems weird if <\nabla f(x_t), g_t> would be a distribution. Figure 5 has only subfigure captions. Experiments of Section 5, legend in Figure 5 (?) says DP-SGD, but is it truly DP version of just clipped gradient SGD? ############################### Post-rebuttal update. Thanks authors for addressing my concerns and the extra experiments on connecting DP-SGD and Sec 5.


Review 2

Summary and Contributions: This paper gives a thorough analysis on the gradient clipping in private SGD. More specifically, it demonstrates how gradient clipping can prevent SGD from converging to stationary point. Also, the results provide an explanation why private SGD with gradient clipping remains effective in practice despite its potential clipping bias. Finally, this paper develops a new perturbation-based technique that can provably correct the clipping bias even for instances with highly asymmetric gradient distributions.

Strengths: This paper addresses an important issue in private SGD: how to understand the gradient clipping. The results explain why the private SGD with gradient clipping remains effective in practice despite its potential clipping bias, from a geometric perspective. Also, clipping bias correction approach is proposed via a perturbation-based technique.

Weaknesses: Experimental results should be given to support the Theorems in Section 2.

Correctness: Yes

Clarity: Yes, very well-written.

Relation to Prior Work: Yes

Reproducibility: Yes

Additional Feedback:


Review 3

Summary and Contributions: The paper analyzes the DPSGD algorithm and provides a variant with better provable guarantee.

Strengths: The paper theoretically analyzes the DPSGD algorithm and provides a stronger theoretical guarantee compared to existing results. The analysis depends on introducing an auxiliary distribution that is close to the existing gradient noise distribution. The authors also propose a variant of the DPSGD algorithm that has better provable guarantee.

Weaknesses: The results depend on how close the underlying noise distribution of the stochastic gradients is to a symmetric distribution, tilde{p}. Therefore most of the results have a dependence on tilde{p}, which is hard to interpret or verify in practice. The assumption that the noise distribution is close to a symmetric distribution may not hold for many neural networks with typically sparse gradient updates such as LSTM models. ---- Post rebuttal edit: Thanks for answering my questions. I have read the rebuttal and will keep my score. Minor comment:  Figure 1 doesn't seem to be referred anywhere in the text.

Correctness: Yes

Clarity: Yes. There are few typos: line 132: sqaured --> squared line 157: space is missing before the period.

Relation to Prior Work: Yes

Reproducibility: Yes

Additional Feedback:


Review 4

Summary and Contributions: This paper analyses gradient clipping and how it biases convergence to the optimum. It goes on to analyze differentially private gradient clipping. Bounds are given in terms of a bias term, which translates to a distance measure of the distribution of gradient noise to a symmetric distribution (or mixture). The bias term itself is not bounded, which makes the other bounds less meaningful. However the bias term is analyzed for a few classes of gradient distributions. Adding Gaussian noise before clipping does allow one to bound the bias (but now there are no differentially private guarantees).

Strengths: Significance - Understanding gradient clipping is an important problem, as gradient clipping is extensively used in practice - Bounding the bias in Theorem 6 and illustrating its performance for different values of k in Section 5 clearly demonstrates the benefit of the bounds and better understanding how clipping effects optimization Novelty - The use of symmetric noise distributions to give bounds for clipping is, to my knowledge, new. Relevance - The work should be relevant to both theoretical and empirical practitioners in the Neurips community.

Weaknesses: Significance - I don't find many of the bounds particularly useful, since the bias term itself is not bounded. In section 5 a bound is given for the bias, which is great, but then you have to add noise to the gradients, which is not particularly interesting as it applies neither to DP or non-DP clipping. Also, there is a complicated dependence on k in two terms. The work would be stronger if there were general bounds for the bias, or if the method in section 5 was shown to also be differentially private. Novelty - Clipping has been studied a fair amount in the literature. I don't know how different the conclusions of this paper are from previous work. Other comments - Line 106-107: The smoothness assumption is not valid for neural networks with relu activation functions. This limitis the applicability of the work. - The symmetry assumptions would probably not hold for neural networks which have exploding gradients, like RNNs. - In the Experiment section 4, instead of showing random projections, it would be more information to show projections that had the least symmetry. This could be done by sampling a large number of random directions and then only displaying a few with the least symmetry.

Correctness: I did not check carefully, but they seemed correct.

Clarity: In general the paper is well written. There are many minor errors though, that should be corrected: - Line 7-8: "to stationary point" -> "to a stationary point" - Equation (1): "max" -> "min" - Line 68-69: "inner product upper bounds a constant re-scaling". What inner product? - Line 132: "sqaured" -> "squared" - Line 144: "treshold" -> "threshold" - Line 157: Start "Note..." on a new line. Make sense to have a paragraph break here. - Line 253: "seem" -> "seen"

Relation to Prior Work: Some related work is mentioned, but this section could be improved. It is not clear how the conclusions of this paper differ from previous work (e.g. how do the biases calculated in this paper differ from biases in other papers and why is this formulation of the bias better?)

Reproducibility: Yes

Additional Feedback:

[Author Response · NeurIPS 2020]

We thank the reviewers for their positive and constructive feedback. We address several points in the review below.

**Connecting Section 5 to DP-SGD.** The bias reduction technique in Section 5 is designed for DP-SGD with clipping.

When it is applied to DP-SGD, the update rule is shown below.

$$x_{t+1} = x_t - \alpha\bigg(\bigg(\frac{1}{|S_t|}\sum_{i \in S_t} \text{clip}(\nabla f(x_t) + \xi_{t,i} + k\zeta_{t,i}, c)\bigg) + Z_t\bigg),$$

where $\zeta_{t,i} \sim \mathcal{N}(0, I)$ is the noise added to reduce clipping bias and $Z_t \sim \mathcal{N}(0, \sigma^2 I)$ is the noise added for privacy (see
(8) in the paper for other notations). The privacy guarantee of this algorithm is the same as original DP-SGD since the
noise added before clipping does not scale with the corresponding $L_2$ sensitivity and does not provide formal privacy
guarantee. We also compared original DP-SGD and the above algorithm with $\sigma = 1$ in the 3 experiment settings in
Section 5, the results are similar to thoise in the figure in Section 5 (which used $\sigma = 0$ for all algorithms). We will
include the experiments for $\sigma = 1$ in the paper.

**Typos:** Thank you for pointing them out, we will correct the typos.

**Reviewer 1.** *Q1: Discussion on the convergence rate of DP-SGD.* This is indeed a very important point to discuss
in the paper. DP-SGD achieves convergence rate of $O(\sqrt{d}/(n\epsilon))$ in the existing literature. As shown in Theorem 5,
with gradient clipping, the rate becomes $O(\sqrt{d}/(n\epsilon))$ + clipping bias. When gradient distribution is symmetric, the
convergence rate can recover to $O(\sqrt{d}/(n\epsilon))$. We will add a more detailed discussion around Theorem 5.

*Q2: Better connect the clipping bias reduction method with DP-SGD.* Please see "Connecting Section 5 to DP-SGD".

*Q3: Additional feedback.* Thank you for the suggestions, we will revise accordingly.

**Reviewer 2.** *Q1: Experiments to support the Theorems in Section 2.* We will add such experiments in the paper.
Theorem 1 is a standard convergence bound for optimization; we will verify it on MNIST. Theorem 2 is our new
contribution, we conducted some new experiments for it. We considered a 1-dimensional example and choose a
symmetric noise $\xi_t \sim \mathcal{N}(0, 1)$ and set $c = 1$. Then we compare $\mathbb{E}_{\xi_t \sim \tilde{p}}[\langle \nabla f(x_t), g_t \rangle]$ (estimated by averaging $10^5$
samples) with the lower bound in Theorem 2 for different $\|\nabla f(x_t)\|$. The results below verify our lower bound.

| $\|\nabla f(x_t)\|$ | 0.05 | 0.1 | 1 | 2 | 10 | 100 |
|---|---|---|---|---|---|---|
| $\mathbb{E}_{\xi_t \sim \tilde{p}}[\langle \nabla f(x_t), g_t \rangle]$ | 1.7e-4 | 6.6e-3 | 0.612 | 1.83 | 10 | 100 |
| lower bound | 4e-5 | 2e-3 | 0.148 | 0.3 | 1.48 | 14.8 |

**Reviewer 3.** *Q1: Interpretation and verification about $\tilde{p}$ in practice.* We will provide a few examples to illustrate good
choices of $\tilde{p}$ and add more discussion on what is the desired property for $\tilde{p}$. Essentially, we want to choose $\tilde{p}$ such that
the Wasserstein distance term in Theorem 5 is small and the probability term in Theorem 5 is large. For verification of
symmetricity, we found the random projection method used in Figure 1 and 2 to be quite effective in practice. It is
interesting to investigate better ways to measure symmetricity for very high-dimensional distributions.

*Q2: The symmetric distribution may not hold for neural nets such as LSTM.* This is an interesting point. In general, it
will be very interesting to investigate the geometric structure among the gradients for different types of neural network,
even absent privacy or clipping concerns. We will put this in our future work discussion.

**Reviewer 4.** *Q1: Unbounded bias terms make the bounds less useful.* The bias terms in the bounds are bounded
by constants in worst case, and could be 0 for symmetric distributions. For example, in Theorem 5, the clipping
bias is given by the Wasserstein distance term. For iteration $t$, $W_{\nabla f(x_t),c}(\tilde{p}_t, p_t) \le 2\|\nabla f(x_t)\|c$ always holds. More
importantly, $W_{\nabla f(x_t),c}(\tilde{p}_t, p_t) = 0$ if $\tilde{p}_t = p_t$ which can happen when $p_t$ (gradient noise distribution) is symmetric.

*Q2: Application of the technique in Section 5 to DP-clipping.* Please see Section "Connecting section 5 to DP-SGD".

*Q3: How different the conclusions of this paper are from previous work.* As noticed by the reviewer, the use of
symmetric noise distributions in analysis for clipping is new. The key conclusion of our paper is the 0 (small) clipping
bias for symmetric (approximate symmetric) gradient noise distributions. We also proposed a trick to symmetrify
gradient noise distributions to reduce clipping bias. The most related previous works are Pichapati et al. [2019] and
Zhang et al. [2019]. Both works give bounds on clipping bias essentially similar to $c\|\nabla f(x_t)\|\mathbb{P}[\text{gradient is clipped}]$
which could be large when the clipping probability is large. Compare with these works, we give sharper bounds for
(approximate) symmetric gradient noise distributions. Another closely related work is [1], which was uploaded to arxiv
after the submission deadline. The work proves that clipping can lead to constant regret for DP-SGD, like in Example 1
in our paper. We will add a detailed discussion on comparison with these works.

*Q4: Smoothness assumption limits the applicability, symmetricity assumption may not hold for RNNs, show least
symmetric projections.* Smooth assumption is commonly used for DP-SGD in literature and it holds for smooth
approximations for relu such as softplus and swish. Extending the analysis of DP-SGD to non-smooth problems is
important and interesting even absent clipping. The gradient distribution for RNNs is also a very interesting point to
explore. We will mention these in our future work discussion. We will add figures for projections with least symmetry.

[1] Shuang Song, Om Thakkar, Abhradeep Thakurta. Characterizing private clipped gradient descent on convex
generalized linear problems. *arXiv preprint arXiv:2006.06783, 2020.*

[Meta-Review · NeurIPS 2020]

Although there is still room for improvement (e.g., it's unclear that the dependence on the symmetry of the gradient distribution is fundamental), overall the reviewers agreed that this paper is interesting, well-written, and provides interesting new insights into an important set of algorithms.